# Hydroclimatic variations in southeastern China during the 4.2 ka event reflected by stalagmite records

Haiwei Zhang[1,2], Hai Cheng[1,3], Yanjun Cai[2,1,6], Christoph Spötl[4], Gayatri Kathayat[1], Ashish Sinha[5], R. Lawrence Edwards[3], and Liangcheng Tan[2,1,6]

[1]Institute of Global Environmental Change, Xi'an Jiaotong University, Xi'an 710054, China

[2]Institute of Earth Environment, Chinese Academy of Sciences, State Key Laboratory of Loess and Quaternary Geology, Xi'an 710061, China

[3]Department of Earth Sciences, University of Minnesota, Minneapolis, Minnesota 55455, USA

[4]Institute of Geology, University of Innsbruck, Innsbruck 6020, Austria

[5]Department of Earth Science, California State University Dominguez Hills, Carson, California 90747, USA

[6]Open Studio for Oceanic-Continental Climate and Environment Changes, Pilot National Laboratory for Marine Science and Technology (Qingdao), Qingdao 266061, China

*Corresponding to*: Haiwei Zhang (zhanghaiwei@xjtu.edu.cn)

**Abstract.** Although the collapses of several Neolithic cultures in China are considered to have been associated with abrupt climate change during the 4.2 ka BP event (4.2-3.9 ka BP), the timing and nature of this event and the spatial distribution of precipitation between northern and southern China are still controversial. Until now, the hydroclimate of this event in southeastern China is still poorly known, except for a few published records from the lower reaches of Yangtze River. In this study, a high-resolution record of monsoon precipitation between 5.3 and 3.57 ka BP based on a stalagmite from Shennong Cave, Jiangxi Province, southeast China, is presented. Coherent variations in $\delta^{18}O$ and $\delta^{13}C$ reveal that the climate in this part of China was dominantly wet between 5.3 and 4.5 ka BP and mostly dry between 4.5 and 3.57 ka BP, interrupted by a wet interval (4.2-3.9 ka BP). A comparison with other records from monsoonal China suggests that summer monsoon precipitation decreased in northern China but increased in southern China during the 4.2 ka BP event. We propose that the weakened East Asian summer monsoon controlled by the reduced Atlantic Meridional Overturning Circulation resulted in this contrasting distribution of monsoon precipitation between northern and southern China. During the 4.2 ka BP event the rain belt remained longer at its southern position, giving rise to a pronounced humidity gradient between northern and southern China.

**1 Introduction**

The 4.2 ka BP event was a pronounced climate event in the Holocene which has been widely studied in the past 20 years. It was identified as an abrupt (mega)drought and/or cooling event in a variety of natural archives including ice cores, speleothems, lake sediments, marine sediments, and loess. This climate episode was associated with the collapse of several ancient civilizations and human migrations in many sites worldwide (e.g., Egypt, Greece, the Indus Valley and the Yangtze Valley) (Weiss et al., 1993; Cullen et al., 2000; Gasse, 2000; DeMenocal, 2001; Weiss and Bradley, 2001; Thompson et al., 2002; Booth et al., 2005; Bar-Matthews and Ayalon, 2011; Berkelhammer et al., 2012; Ruan et al., 2016; Railsback et al., 2018). Recently, the 4.2 ka BP event was defined as the lower boundary of the Meghalayan Stage by the International Commission on Quaternary Stratigraphy. The timing of this geological boundary was defined at a specific level (i.e., the transformation from calcite to aragonite accompanied by an abrupt increase in $\delta^{18}O$) in a stalagmite from northeast India (Walker et al., 2018).

The abrupt climate change associated with the 4.2 ka BP event has been proposed to have contributed to the collapses of Neolithic cultures in China (Jin and Liu, 2002; Huang et al., 2010, 2011; Zhang et al., 2010; Liu and Feng, 2012; Wu et al., 2017). Most of these studies imply a temperature drop in continental China at about 4.2 ka BP (Yao and Thompson, 1992; Jin and Liu, 2002; Zhou et al., 2002; Xu et al., 2006; Yao et al., 2017; Zhao et al., 2017), but changes in the spatial distribution of precipitation are also discussed (Tan et al., 2008; Huang et al., 2010, 2011; Tan et al., 2018a, 2018b; Wu et al., 2017). For example, a grain-size record from Daihai Lake, north China, suggests a decrease in monsoon precipitation between 4.4 and 3.1 ka BP with a very dry interval between 4.4 and 4.2 ka BP (Peng et al., 2005). Extreme flooding during the 4.2 ka BP event was identified by paleoflood deposits in the middle reaches of the Yellow River (Huang et al., 2010, 2011). Wu et al. (2017) reported evidence of two extraordinary paleoflood events in the middle reaches of the Yangtze River at 4.9-4.6 ka BP and 4.1-3.8 ka BP, closely related to the expansion of the Jianghan lakes. These extreme hydroclimate events may have accelerated the collapse of the Shijiahe Culture in the middle reaches of the Yangtze River (Wu et al., 2017). Multiple proxies in four stalagmites from Xianglong Cave, south of the Qinling Mountains, indicate that the upper Hanjiang River region experienced a wet climate during the 4.2 ka BP event (Tan et al., 2018a). Peat records provide a broad picture of climate variations in southeast China (SEC) during the Holocene (Zhou et al., 2004; Zhong et al., 2010a, 2010b, 2010c, 2015, 2017). The resolution of these records, however, is not high enough to study the detailed structure of the 4.2 ka BP event. Until now, there is only one published stalagmite record from SEC (Xiangshui Cave; Fig. 1), indicating a wet interval during the 4.2 ka BP event (Zhang et al., 2004).

The aim of this work was to obtain a high-resolution stalagmite-based record from SEC to

study the hydroclimatic variations during the 4.2 ka BP event and to compare them to records in northern China in order to explore the possible north-south precipitation gradient during this event.

**2 Study area and sample**

Shennong Cave (28°42' N, 117°15' E, 383 m a.s.l.) is located in the northeastern Jiangxi Province, SEC (Fig. 1), a mid-subtropical region strongly influenced by the East Asian summer monsoon (EASM). Mean annual precipitation and temperature at the nearest meteorological station (Guixi station; 1951-2010 AD) are 1857 mm and 18.5 °C, respectively (Fig. 2a). Shennong Cave is located in the region of the spring persistent rain. The rainy season includes both summertime monsoon rainfall and spring persistent rain (Tian and Yasunari, 1998; Wan et al., 2008; Zhang et al., 2018). The latter is a unique synoptic and climatic phenomenon that occurs from March until mid-May, mostly south of the Yangtze River (about 24°N to 30°N, 110°E to 120°E - Tian and Yasunari, 1998; Wan and Wu, 2007, 2009). EASM precipitation lasts from mid-May to September (Wang and Lin, 2002). In the region of spring persistent rain, the EASM (May to September) precipitation accounts for 54% of the annual precipitation and the non-summer monsoon (NSM, October to next April) precipitation accounts for 46% (Zhang et al., 2018). The distribution of EASM vs. NSM precipitation amount in this region is distinctly different from that in the northern and southwestern part of monsoonal China, where the mean annual percentage of EASM (65-90%) is much higher than the mean annual percentage of NSM (10-35%). Data from the nearest GNIP station in Changsha, also located in the region of the spring persistent rain, indicate that the $\delta^{18}O$ values of EASM precipitation are lower comparing with those of NSM precipitation (Fig. 2b). Two years monitoring data (2011-2013) in Shennong Cave indicate that the speleothem $\delta^{18}O$ values reflected drip water $\delta^{18}O$ values inherited from the amount-weighted annual precipitation $\delta^{18}O$ outside the cave. Therefore, different from the southwestern and northern part of the monsoonal China, where the speleothem $\delta^{18}O$ values are mainly influenced by EASM precipitation, speleothem $\delta^{18}O$ from the Shennong Cave is controlled by both EASM and NSM precipitation (Zhang et al., 2018).

The cave developed in Carboniferous limestone of the Chuan-shan and Huang-long Groups, which are mainly composed of limestone and interbedded dolostone. The thickness of the cave roof ranges from about 20 to about 80 m, with an average of ~50 m. The overlying vegetation consists mainly of secondary forest tree species such as *Pinus*, *Cunninghamia* and *Phyllostachys* and shrub-like *Camellia oleifera* and *Ilex* which are $C_3$ plants (Zhang et al., 2015). Two years monitoring data show that the mean temperature in the cave is 19.1 °C with a standard deviation of 2.5 °C (Fig. 2c), consistent with mean annual air temperature outside the cave (Fig. 2a). The relative humidity in the interior of the cave approaches 100% during the most of the year (Fig. 2c). Abundant aragonite and calcite speleothems are present in the cave. Their mineralogy is likely controlled by the Mg/Ca ratio

of the drip water reflecting the variable dolomite content of the host rock (De Choudens-Sanchez and Gonzalez, 2009; Zhang et al., 2014, 2015). All aragonite stalagmites were deposited within ~1.5 km of the cave entrance where the bedrock is dolomite and all calcite stalagmites were deposited in more distal parts of the cave where limestone constitutes the host rock (Zhang et al., 2015).

In November 2009 stalagmite SN17 (Fig. 3), 320 mm in length, was collected 200 m behind the cave entrance where the bedrock is dolomite. X-ray diffraction (XRD) analyses suggest that the stalagmite is composed of aragonite, except for the bottom section below 318 mm which is composed of calcite (Fig. 3). The calcite section was not included in the present study.

**3 Methods**

**3.1 $^{230}$Th dating**

        Fifteen subsamples for $^{230}$Th dating were drilled along the growth axis of SN17 (Fig. 3) with a hand-held carbide dental drill, and were dated on a multi-collector inductively coupled plasma mass spectrometer (MC-ICP-MS, Neptune Plus) at the Department of Earth Sciences, University of
Minnesota and the Institute of Global Environmental Change, Xi'an Jiaotong University (Cheng et al., 2000, 2013). The chemical procedure used to separate uranium and thorium followed those described by Edwards et al. (1987).

        **3.2 Stable isotope analyses**

120 sub-samples for stable isotope analyses were drilled along the central axis of SN17 at intervals of 1 mm between 0 and 70 mm and 5 mm between 70 and 320 mm distance from top, respectively. The samples were analysed using a gas-source stable isotope ratio mass spectrometer (Isoprime100), equipped with a MultiPrep system at the Institute of Earth Environment, Chinese Academy of Sciences (IEECAS). The international standard NBS19 and the laboratory standard HN
were analysed after every 10 sub-samples to monitor data reproducibility. All oxygen and carbon isotope compositions are reported in per mil relative to the Vienna Pee Dee Belemnite (VPDB). Reproducibility of $\delta^{18}$O and $\delta^{13}$C values was better than 0.1‰ and 0.08‰ (2σ), respectively.

        **4 Results**

**4.1 Chronology**

        The $^{230}$Th dates are all in stratigraphic order and no significant hiatus was observed (Table 1 and Fig. 3). Because of the high uranium concentrations (1120-6380 ppb) and relative low thorium concentrations (56-195 ppt), most of the dating errors are less than 6‰. We used COPRA (Breitenbach et al., 2012) to establish an age model of stalagmite SN17 which grew from 3570 to
5303 a BP (Fig. 3).

## 4.2 δ¹³C and δ¹⁸O records

The temporal resolution of the $\delta^{13}C$ and $\delta^{18}O$ records ranges from 6 to 21 years. The $\delta^{13}C$ values fluctuate around -9.18‰ (mean value) during the period 5.3 to 4.5 ka BP, and increase to -8.69‰ between 4.5 and 3.57 ka BP (Fig. 4a). $\delta^{18}O$ fluctuates around -6.75‰ (mean value) during the period 5.3 to 3.57 ka BP on decadal to centennial timescales (Fig. 4c). The $\delta^{18}O$ record is broadly similar to the $\delta^{13}C$ record between 5.3 and 3.57 ka BP (Figs. 4 and 6c), with a significantly positive correlation (r=0.37, p<0.01; Fig. 5c). Both $\delta^{13}C$ and $\delta^{18}O$ records were normalized to the standard z-score (Figs. 4b and d), which clearly shows the decadal to centennial variability during the interval of 5.3-3.57 ka BP. Z-scored $\delta^{13}C$ values are above 0 during the intervals of 5.3-4.5 and 4.15-3.95 ka BP and below 0 during the intervals of 4.5-4.15 and 3.95-3.57 ka BP on centennial timescales (Fig. 4b). Z-scored $\delta^{18}O$ record are above 0 during the intervals of 5.3-5.05, 4.9-4.66, 4.57-4.48 and 4.32-4.06 and below 0 during the intervals of 5.05-4.9, 4.66-4.57, 4.48-4.32 and 4.06-3.57 ka BP on centennial timescales (Fig. 4d). The growth rate of SN17 shows a persistently decreasing trend from 0.62 to 0.034 mm/yr between 5.3 and 4.5 ka BP, followed by low values during the period 4.5 to 3.57 ka BP with relatively higher values between 4.26 and 4.0 ka BP (Fig. 4e). The periods of higher growth rate correspond to the periods of lower $\delta^{18}O$ and $\delta^{13}C$ values, which we infer to be the time of more summer monsoon precipitation (see below).

## 5 Discussion

### 5.1 Test of equilibrium deposition

Speleothem $\delta^{18}O$ can be used to indicate climatic variation provided that the speleothem was precipitated at or close to isotopic equilibrium. The "Hendy test" is a widely used approach to explore to which extent calcite deposition on the stalagmite surface occurred in isotopic equilibrium with the parent drip water. Following Hendy (1971), twenty-one subsamples from three growth layers were analyzed, and no progressive increase in $\delta^{18}O$ along individual growth layers and no significant correlation of coeval $\delta^{18}O$ and $\delta^{13}C$ values were found (Figs. 5a and b). This suggests (albeit not proves) that the stalagmite SN17 was deposited close to isotopic equilibrium. On the other hand, $\delta^{18}O$ and $\delta^{13}C$ values show a statistically significant covariance along the growth axis (r=0.37, p<0.01; Fig. 5c) suggesting that the speleothem might be effected by kinetic fractionation (Dorale and Liu, 2009). Some studies, however, demonstrated that stalagmites showing a significant correlation between $\delta^{18}O$ and $\delta^{13}C$ can also have formed under isotopic equilibrium, if both parameters are controlled by the common factors (Dorale et al., 1998; Dorale and Liu, 2009; Tan et

al., 2018a). A more robust test is the replication of $\delta^{18}O$ records from different caves (Dorale et al.,

1998; Wang et al., 2001; Dorale and Liu, 2009; Cai et al., 2010). The $\delta^{18}O$ records of SN17 and

those from Dongge (Wang et al., 2005) and Xiangshui Caves (Zhang et al., 2004), located southwest

of Shennong Cave (Fig. 1), show remarkable similarities in the overlapping interval (Figs. 6a and

b). The replication of these records further confirms that aragonite of stalagmite SN17 was most

likely deposited close to isotopic equilibrium, i.e., its $\delta^{18}O$ variations primarily reflect climatic

changes.

## 5.2 Interpretation of $\delta^{18}O$ and $\delta^{13}C$

The climatic significance of the speleothem $\delta^{18}O$ from the monsoonal China has been

intensively debated in recent years. Most scientists agree that speleothem $\delta^{18}O$ in the monsoonal

China represent variations in EASM intensity and/or changes in spatially-integrated precipitation

between different moisture sources and the cave site on millennial timescales (Cheng et al., 2016).

Some researchers, however, suggest that the Chinese speleothem $\delta^{18}O$ is influenced by moisture

circulation on interannual to centennial timescales (Tan, 2009, 2014, 2016). In the region of spring

persistent rain, the speleothem $\delta^{18}O$ values from Shennong Cave are controlled by both EASM and

NSM precipitation. A 200-year speleothem $\delta^{18}O$ record from E'mei Cave, located 160 km northwest

of Shennong Cave, shows a significantly negative correlation with both EASM precipitation amount

(r=-0.54, p<0.01) and the EASM/NSM ratio (r=-0.67, p<0.01) during 1951-2009 AD (Fig. 3 in

Zhang et al., 2018). The EASM precipitation amount varies in a same direction as the EASM/NSM

ratio, i.e., an increasing (decreasing) EASM/NSM ratio corresponds to more (less) EASM

precipitation. In addition, the E'mei $\delta^{18}O$ record also exhibits a coherent variation with the

drought/flood index during 1810-2010 AD on decadal to centennial timescales (Fig. 2 in Zhang et

al., 2018). This indicates that E'mei $\delta^{18}O$ is likely dominated by the EASM precipitation amount on

decadal to centennial timescales. Therefore, we suggest that the stalagmite $\delta^{18}O$ record from

Shennong Cave, similar to E'mei Cave, might be primarily influenced by the EASM/NSM ratio and

also affected by the EASM precipitation amount on interannual to decadal timescales, and can be

dominated by the EASM precipitation amount on decadal to centennial timescales, i.e., lower

(higher) $\delta^{18}O$ values corresponding to higher (lower) EASM/NSM ratios and more (less) EASM

precipitation.

Changes in speleothem $\delta^{13}C$ are generally controlled by vegetation density and composition in

the catchment which vary according to the hydroclimate (Genty et al., 2001, 2003, 2006; McDermott,

2004; Baldini et al., 2005; Cruz Jr et al., 2006; Fairchild et al., 2006; Fleitmann et al., 2009; Noronha

et al., 2015; Wong and Breeker, 2015). In regions where the vegetation type is predominantly $C_3$ or

$C_4$ plants, a dry climate will lead to a reduction of the vegetation cover, density and soil microbial activity as well as an increase in the groundwater residence time allowing more $\delta^{13}C$-enriched

bedrock to be dissolved. In addition, prior calcite precipitation (PCP) in the vadose zone will result in higher $\delta^{13}C$ values accompanied by increased Mg/Ca ratios in speleothems (Baker et al., 1997). Slow drip rates and increased evaporation and/or ventilation inside the cave will lead to higher $\delta^{13}C$ values, usually accompanied by kinetic isotopic fractionation (Fairchild et al., 2000; Oster et al., 2010; Frisia et al., 2011; Li et al., 2011; Tremaine et al., 2011; Meyer et al., 2014). In Jiangxi

Province, a region presently occupied by mostly $C_3$ plants, no evidence has been found for a replacement of $C_3$ plants by $C_4$ plants during the mid- to late Holocene (Zhou et al., 2004; Zhong et al., 2010b). There is no significantly positive correlation between $\delta^{13}C$ values and Mg/Ca ratios between 5.3 and 3.57 ka BP (Fig. 5d). In Shennong Cave ventilation is weak and relative humidity remains close to 100% throughout the year. Rapid $CO_2$ degassing is less common under these

conditions. Stalagmite SN17 was likely deposited close to isotopic equilibrium, as confirmed by the "Hendy test" and the "replication test" (section 5.1). The $\delta^{13}C$ variations in this stalagmite were primarily driven by vegetation density and soil bioproductivity associated with hydroclimatic variations but not by PCP or rapid $CO_2$ degassing, with lower $\delta^{13}C$ values corresponding to a denser vegetation cover associated with a wet climate, and vice versa (Zhang et al., 2015).


## 5.3 Hydroclimate between 5.3 and 3.57 ka BP

Previous speleothem studies from monsoonal China suggested a coherent trend of decreasing precipitation from the early to the late Holocene (Wang et al., 2005; Hu et al., 2008; Cai et al., 2010, 2012; Dong et al., 2010, 2015; Jiang et al., 2013; Tan et al., 2018a), which follows the gradually

decreasing Northern Hemisphere summer insolation. Our $\delta^{13}C$ record exhibits a similarly increasing trend (Figs. 4a, b), indicating that the climate in our study area changed from wetter to drier condition between 5.3 and 3.57 ka BP. Although the growth rate of stalagmites is often not a direct function of precipitation amount (Railsback, 2018), some studies suggest that in monsoonal China changes in growth rate of stalagmite can be influenced by variations in monsoon precipitation (Wang

et al., 2005). The long-term decreasing trend in growth rate, broadly consistent with changes in the $\delta^{13}C$ record, is possible related to decreased monsoon precipitation between 5.3 and 3.57 ka BP (Fig. 4). But it should be noted that more monitoring data are need to confirm this relationship between growth rate and precipitation in our study cave. The long-term trend in $\delta^{18}O$ is less significant than those of $\delta^{13}C$ and growth rate (Fig. 4) and might be caused by changes in precipitation seasonality

since the mid-Holocene, i.e., variations in the EASM/NSM ratio. In this paper, we focus on the timing and nature of the 4.2 ka BP event. The long-term trend of $\delta^{18}O$ is not discussed in detail, because it remains unclear how the EASM and NSM precipitation varied during the Holocene.

As discussed in section 5.2, the SN17 $\delta^{18}O$ might be dominated by EASM precipitation amount on decadal to centennial timescales, although the EASM/NSM ratio has a significant impact on interannual to decadal timescales. On decadal to centennial timescales the SN17 $\delta^{18}O$ record shows a coherent variability with the $\delta^{13}C$ record (Fig. 6c), which is primarily influenced by EASM precipitation amount but not by seasonal precipitation $\delta^{18}O$. In addition, the SN17 $\delta^{18}O$ record is remarkably similar to the $\delta^{18}O$ record from Dongge Cave (Fig. 6a), which is dominated by summer monsoon precipitation (Wang et al., 2005). These observations indicate that, on decadal to centennial timescales, variations in SN17 $\delta^{18}O$ might primarily reflect changes in EASM precipitation amount, although the EASM/NSM ratio may also have an impact, with higher (lower) $\delta^{18}O$ values corresponding to decreased (increased) summer monsoon precipitation. The asynchronous variations between the $\delta^{13}C$ and $\delta^{18}O$ during the short intervals of 4.7-4.6, 4.3-4.2 and 4.05-3.95 ka BP (Fig. 6c) may be ascribed to the delayed response of vegetation density to the variations in EASM precipitation amount or the EASM/NSM ratio. During 4.5-3.57 ka BP, a wet interval between 4.2 and 3.9 ka BP can be identified in both $\delta^{18}O$ and $\delta^{13}C$ records, consistent with the time of high growth rate between 4.26 and 4.0 ka BP (Figs. 4 and 6c).

Therefore, we suggest that the climate in the study area between 5.3 and 4.5 ka BP was dominantly wet, and changed to a rather dry climate between 4.5 and 3.57 ka BP, interrupted by one wet interval between 4.2 and 3.9 ka BP (Figs. 4 and 6c). It indicates that the climate in our study area during the 4.2 ka BP event (4.2-3.9 ka BP) was predominantly wet.

**5.4 Comparison with other records in monsoonal China covering the 4.2 ka event**

The nature and timing of the 4.2 ka BP event in southern and northern China are still controversial, because the discrepancies might be also caused by the large dating uncertainties and the low proxy resolution in some records. By reviewing records from the monsoon region of China, Tan et al. (2018a) has already proposed a "north-dry and south-wet" pattern during the 4.2 ka BP event, however, more speleothem records from SEC are still needed to confirm this pattern. In this section, we compare the high-precision and high-resolution speleothem records from the monsoonal China during the interval of 5.4-3.6 ka BP.

For northern and southwestern China, the stalagmite $\delta^{18}O$ record from Lianhua Cave (Fig. 1) shows a long-term decreasing trend of summer monsoon precipitation (Fig. 7a, Dong et al., 2015), consistent with both $\delta^{18}O$ and $\delta^{13}C$ records of stalagmites from Nuanhe Cave in northeastern China (Fig.1, Tan and Cai, 2005; Wu et al., 2011; Zhang and Wu, 2012). These records indicate that the climate in northern China gradually varied from wet to dry between 5.4 and 3.6 ka BP, and the climate was very dry during the interval of 4.2-3.9 ka BP. The $\delta^{18}O$ record from Dongge Cave, southwest of the monsoonal China, reveals a dry event between 4.4 and 3.95 ka BP (Fig. 7g, Wang

et al., 2005), which is consistent with a decreased precipitation of Indian summer monsoon during the interval of 4.3-3.9 ka BP (Fig. 7h, Berkelhammer et al., 2012) and dry intervals in the stalagmite records from Shigao, Dark and Xianren Caves, southwestern China (Fig. 8 and references therein), except wet cliamte documented by multiproxy data from a maar lake in southwest China (Fig. 8; Zhang et al., 2017). The prominent drought during the 4.2 ka BP event was also recorded by various other archives from sites in northern and southwestern China (Fig. 8 and references therein).

For north and south of the Qinling Mountains, two stalagmite $\delta^{18}O$ records from Jiuxian and Xianglong Caves south of these mountains exhibit coherent variations on centennial timescales but neither of them shows a long-term increasing trend (Figs. 7b and c; Cai et al., 2010; Tan et al., 2018a). Both $\delta^{18}O$ records reveal increased monsoon precipitation between 4.3 and 3.8 ka BP, indicating that the climate south of the Qinling Mountains was wet during the 4.2 ka BP event (Tan et al., 2018a). It should be noted that there is an asynchronous variation during the interval of 4.2-3.9 ka BP between the Jiuxian and Xianglong $\delta^{18}O$ records, which might be due to either dating uncertainties of the Jiuxian record or the spatial distribution of monsoon precipitation between these two regions. The extraordinary flood during the 4.2 ka BP event was also identified in the middle reaches of the Yellow River north of the Qinling Mountains, north-central China (Fig. 8; Huang et al., 2010, 2011).

For south-central China, two stalagmite $\delta^{18}O$ records from Sanbao (Fig. 7d; Dong et al., 2010) and Heshang (Fig. 7e; Hu et al., 2008) Caves in the middle reaches of Yangtze River, south-central China, also indicate a wet interval between 4.2 and 3.9 ka BP, which is consistent with a $\delta^{13}C$ peat record from the Dajiuhu basin (Ma et al., 2008) and paleoflood sediments from Jianghan Plain (Wu et al., 2017) in the same region (Fig. 8). $\delta^{15}N$ and $\delta^{13}C$ records from the Daping swamp in Hunan Province, south-central China, also reveal a wet interval at 4.5-4.0 ka BP (Fig. 8; Zhong et al., 2017).

For SEC, SN17 $\delta^{18}O$ and $\delta^{13}C$ records reveal a wet interval between 4.2 and 3.9 ka BP on centennial timescales (Figs. 6c and 7f), which is consistent with the speleothem $\delta^{18}O$ record from Xiangshui Cave (Figs. 6b). A record of total organic carbon (TOC) from the Dahu swamp, Jiangxi Province, located 450 km south of Shennong Cave, indicates a dry climate between 6.0 and 4.0 ka BP, with a short-lived wet event at 4.1 ka BP (Fig. 8; Zhou et al., 2004). Subsequent multi-proxy records of several new cores from this site also revealed a prevailingly dry climate between 6.0 and 3.0 ka BP with a wet interval at 4.2-3.9 ka BP (Zhong et al., 2010a, 2010b, 2010c). Pollen data of two sediment profiles from the Daiyunshan Mountain, SEC, indicate a centennial-scale wet event at 4.4 ka BP (Fig. 8; Zhao et al., 2017). These published records from SEC are consistent with the SN17 $\delta^{18}O$ and $\delta^{13}C$ records within error, indicating a wet climate in SEC during the 4.2 ka BP event (Fig. 8).

To sum up, high-resolution stalagmite records indicate that the climate during the 4.2 ka BP

event was dry in northern and southwestern China and wet in southern China (Figs. 7 and 8), and the nature and timing of this event were different in different regions of the monsoonal China. The remarkable dry climate lasted from ~4.4 to 3.9 ka BP in northern and southwestern China (Fig. 8). South-central China and SEC was dry between 4.4 and 4.2 ka BP and was wet between 4.2 and 3.9 ka BP (Figs. 8). The climate was wet between 4.4 and 4.2 ka BP but was dry between 4.2 and 3.9 ka BP at Jiuxian Cave (Figs. 7b and 8), southeast of the Qinling Mountains, and the climate was wet between 4.3 and 3.8 ka BP at Xianglong Cave, southwest of this mountain range (Figs. 7c and 8). A dry and cold period during the 4.2 ka BP event was identified in a sediment profile from Gaochun, west of Taihu Lake, East China (Fig. 8, Yao et al., 2017). Therefore, we suggest that the boundary between the dry north and the wet south during the 4.2 ka BP event probably was located along the northern rim of the Qinling Mountains and the lower reaches of the Yangtze River (Fig. 8).

This south-north distribution of monsoon precipitation might have been caused by a weakened EASM intensity, which could have resulted from a reduced Atlantic Meridional Overturning Circulation (AMOC) recorded by higher abundances of ice-rafted debris (IRD) in the North Atlantic (Fig. 7i). Strong freshwater input into the North Atlantic derived from melting icebergs periodically reduced the AMOC (Bond et al., 2001), causing a temperature decrease in the high northern latitudes and intensified mid-latitude westerlies. As a consequence, the Intertropical Convergence Zone and Northern Hemispheric Westerly jet got stronger and migrated southward, and weakened the Indian summer monsoon and the EASM (Wang et al., 2001; Chiang et al., 2015). The stronger Westerly jet and the weakened EASM delayed the Westerly jet transition from south of the Tibetan Plateau to the north in early-mid May and postponed the onset of the EASM (Chiang et al., 2015; Tan et al., 2018a). The rain belt migrated southward and remained longer in southern China than normal, which reduced rainfall in northern and southwestern China (Figs. 7a and g) but enhanced rainfall in central and southern China (Figs. 7c, d, e and f) during the periods of 4.7-4.5 and 4.2-3.9 ka BP (Zhang et al., 2014).

The SN17 $\delta^{18}O$ record exhibits a coherent variation with the $\delta^{18}O$ record from Dongge Cave within error on centennial timescales but there are some differences in amplitude (Figs. 6a and 7), which can also be found between the Jiuxian and Xianglong records and between the Sanbao and Heshang records (Fig. 7). These discrepancies might be due to chronology offsets, because some records are constrained by two to three dates only. Alternatively, the different amplitudes of these records might be caused by the spatial distribution of monsoon precipitation, reflecting the position and residence time of the rain belt associated with variations in EASM intensity. For example, during the period of 4.2-3.9 ka BP, reduced AMOC caused a dry climate in northern and north-central China but a wet climate in south-central China and SEC (Figs. 7 and 8). During the period of 4.6-4.5 ka BP, however, the records show a dry climate from northern China to south-central China but a wet

climate in SEC (Figs. 7 and 8). Because the AMOC was in a very weak stage at 4.55 ka BP this could have resulted in a weakened EASM and a further southward shift of the monsoonal rain belt,

possibly causing a dry climate in south-central China and a wet climate in SEC. Recently, Yan et al. (2018) used a set of long-term climate simulations and suggested that the 4.2 ka BP event could be related to the slowdown of the AMOC but was more likely caused by internal variability of the climate system. Detailed modeling studies and additional high-resolution records are needed to further investigate the possible causes and mechanisms of this event.


## 6 Conclusions

We reconstructed monsoon precipitation variations in the lower Yangtze River region, SEC, between 5.3 and 3.57 ka BP based on $\delta^{18}O$ and $\delta^{13}C$ records of a precisely dated, high-resolution stalagmite from Shennong Cave in the northern part of Jiangxi Province. The long-term trend of increasing

$\delta^{18}O$ and $\delta^{13}C$ values together a decreasing growth rate is consistent with other stalagmite and peat records from monsoonal China, showing increased monsoon rainfall between 5.3 and 4.5 ka BP and decreased monsoon rainfall between 4.5 and 3.6 ka BP in the study area. A wet episode at 4.2-3.9 ka BP is in agreement with other records from southern China. The boundary between a dry climate in northern China and a coeval wet climate in southern China during the 4.2 ka BP event was

probably located to the north of the Qinling Mountains and the lower reaches of the Yangtze River. This north-south distribution of monsoon precipitation may have been caused by a weakened summer monsoon, which itself may have been related to a reduced AMOC. During a weak summer monsoon, the rain belt remains longer in its southern position, resulting in reduced precipitation in northern China and enhanced precipitation in southern China.


## 7 Author contributions

H.W.Z. designed the research and wrote the first draft of the manuscript. H.C. Y.J.C. C.S. and A.S. revised the manuscript. H.W.Z. and Y.J.C. did the field work and collected the samples. H.W.Z. Y.J.C. and L.C.T. conducted the oxygen isotope measurements. H.W.Z. H.C. G.K. and R.L.E.

conducted the $^{230}$Th dating. All authors discussed the results and provided input on the manuscript.

## 8 Competing interests

The authors declare no competing financial interests.

## 9 Acknowledgments

This study was supported by the NSFC (41502166), NSFC (41731174), NSF (1702816), the China Postdoctoral Science Foundation (2015M580832), the State Key Laboratory of Loess and

Quaternary Geology (SKLLQG1046), the Key Laboratory of Karst Dynamics, Ministry of Land and Resources of the People's Republic of China (MLR) and GZAR (KDL201502), and the Shaanxi

Science Fund for Distinguished Young Scholars (2018JC-023).

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

**Figures and Table**

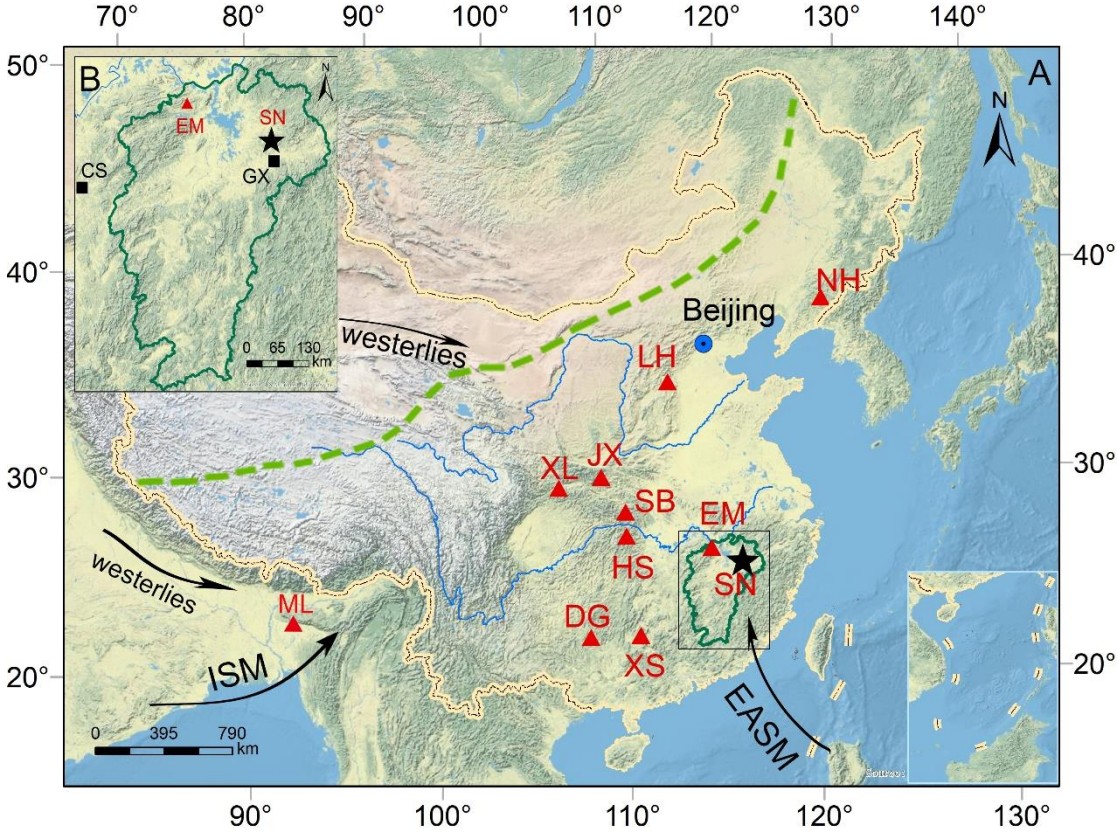

Figure 1. Location of Shennong Cave (SN, black star) and other caves mentioned in the paper. Panel A is an overview topographic map and Jiangxi Province is framed by the green line. Red triangles show the locations of published stalagmite records: NH-Nuanhe Cave (Tan, 2005), LH-Lianhua Cave (Dong et al., 2015), JX-Jiuxian Cave (Cai et al., 2010), XL-Xianglong Cave (Tan et al., 2018a), SB-Sanbao Cave (Dong et al., 2010), HS-Heshang Cave (Hu et al., 2008), EM-E'mei Cave (Zhang et al., 2018), XS-Xiangshui Cave (Zhang et al., 2004), DG-Dongge Cave (Wang et al., 2005) and ML-Mawmluh Cave (Berkelhammer et al., 2012). Black arrows denote the directions of East Asian summer monsoon (EASM), Indian summer monsoon (ISM) and westerlies, which affect the climate in China. Green dash line indicates the limit of modern East Asian summer monsoon. Panel B is an enlarged map showing the locations of Shennong Cave, the Guixi meteorological station (GX) and the GNIP station in Changsha (CS). The base map is the Natural Earth physical map at 1.24 km per pixel for the world (data source: US National Park Service, http://goto.arcgisonline.com/maps/World_Physical_Map. For interpretation of the references to color in this figure legend, the reader is referred to the web version of this article.)

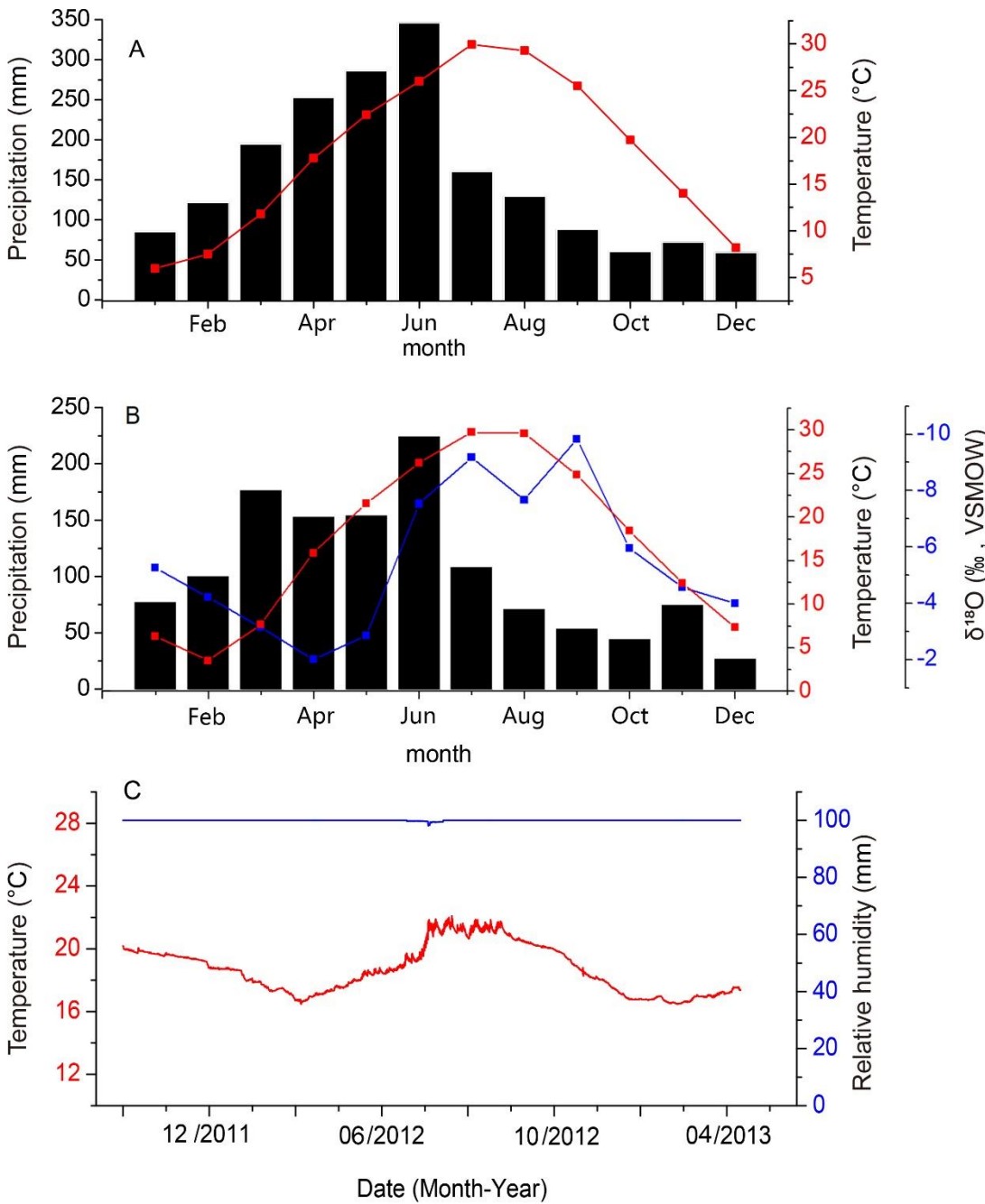


**Figure 2**. Mean monthly temperature, precipitation and δ¹⁸O value from two meteorological stations close to the study area and environmental monitoring in Shennong Cave. (A) Mean monthly air temperature (red line) and precipitation (black column) from the Guixi meteorological station for 1951-2010. (B) Mean monthly air temperature (red line), precipitation (black column) and δ¹⁸O

value (blue line) from the Changsha GNIP station for 1988-1992. (C) Air temperature (red line) and relative humidity (blue line) in Shennong Cave from October 2011 to April 2013.

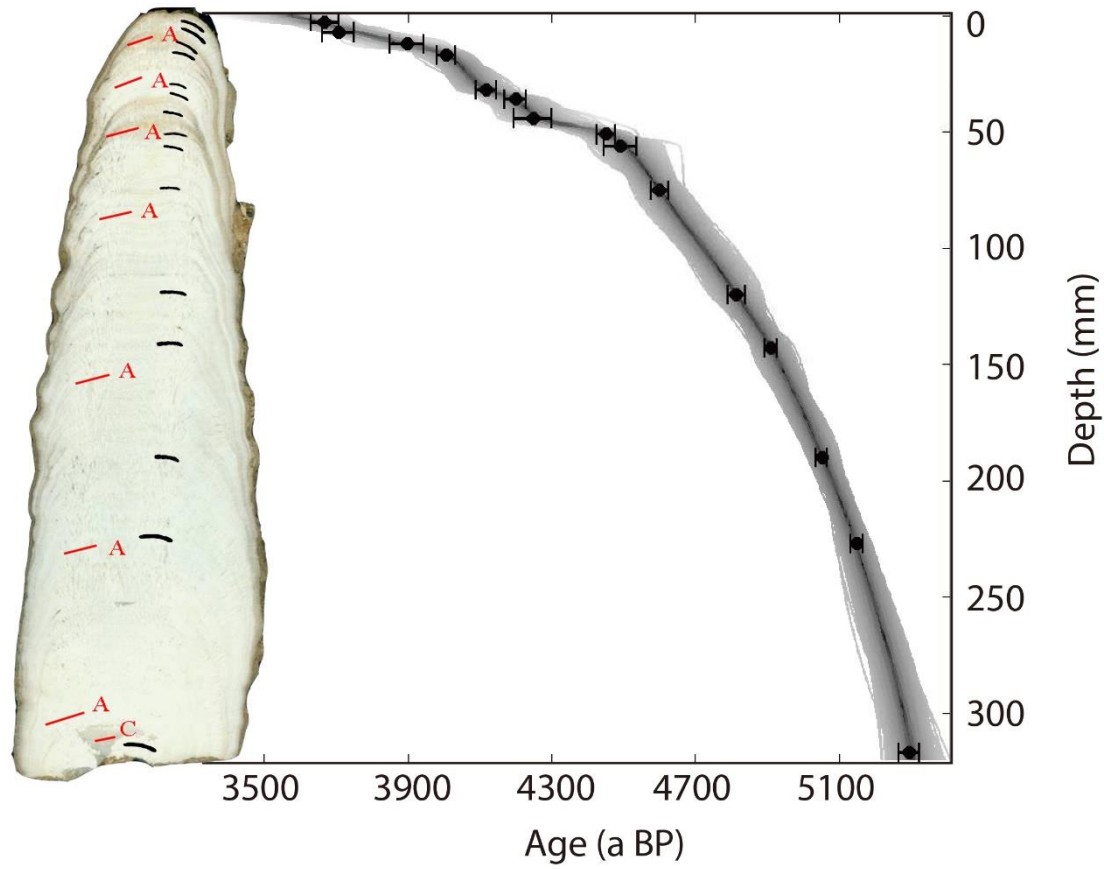

**Figure 3**. Polished section (left) and age model (right) of stalagmite SN17. Sampling positions for
XRD analyses (red lines; A aragonite, C calcite) and [230]Th datings (black lines) are shown on the
slab. SN17 age model and modeled age uncertainties using COPRA, error bars on [230]Th dates
indicate 2σ analytical errors. The gray band depicts the 95% confidence interval.

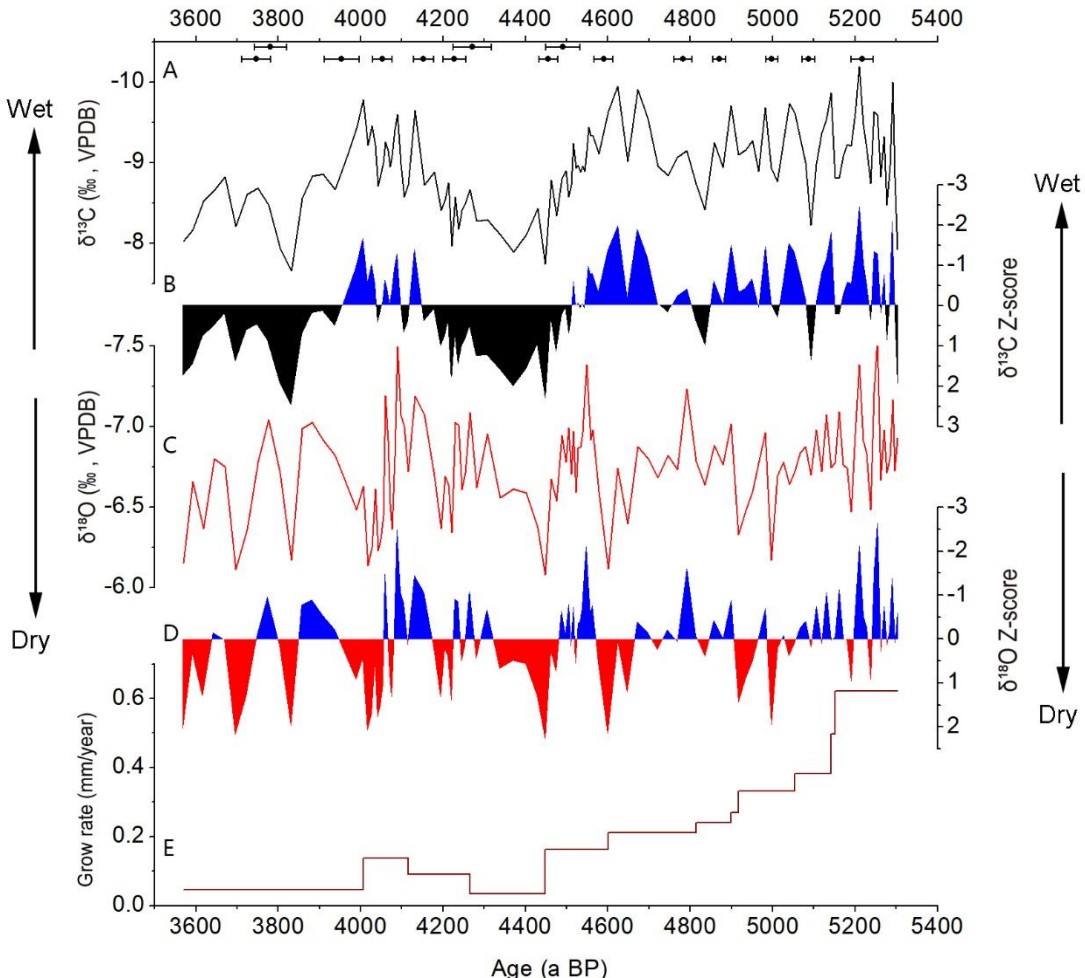

**Figure 4**. δ¹³C (A) and δ¹⁸O (C) records and growth rate (E) of stalagmite SN17. The δ¹³C and δ¹⁸O records were normalized to standard records of z-scored δ¹³C (B) and z-scored δ¹⁸O (D) for comparison, respectively. ²³⁰Th dates and error bars are shown on the top of panel A.

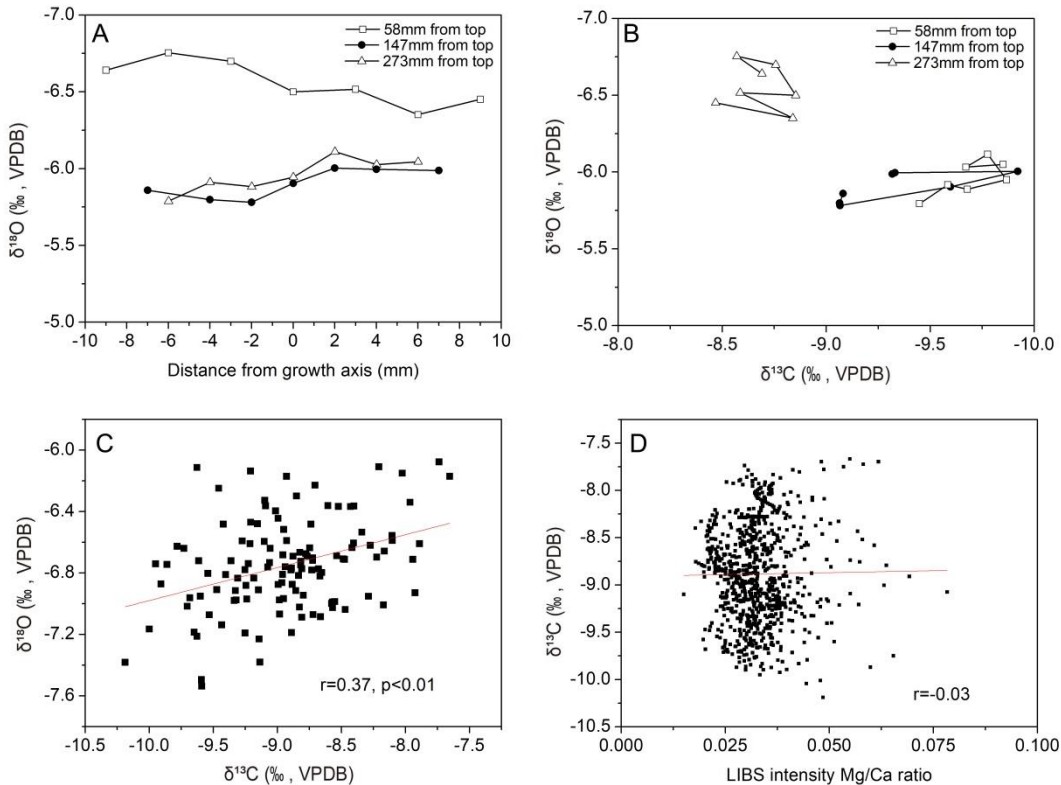

**Figure 5.** Hendy tests of stalagmite SN17 (coeval $\delta^{18}O$ data (A) and $\delta^{18}O$ versus $\delta^{13}C$ plot (B)) and correlation between $\delta^{13}C$ and $\delta^{18}O$ values (C) and between $\delta^{13}C$ values and Mg/Ca ratios (D) measured along the stalagmite growth axis.

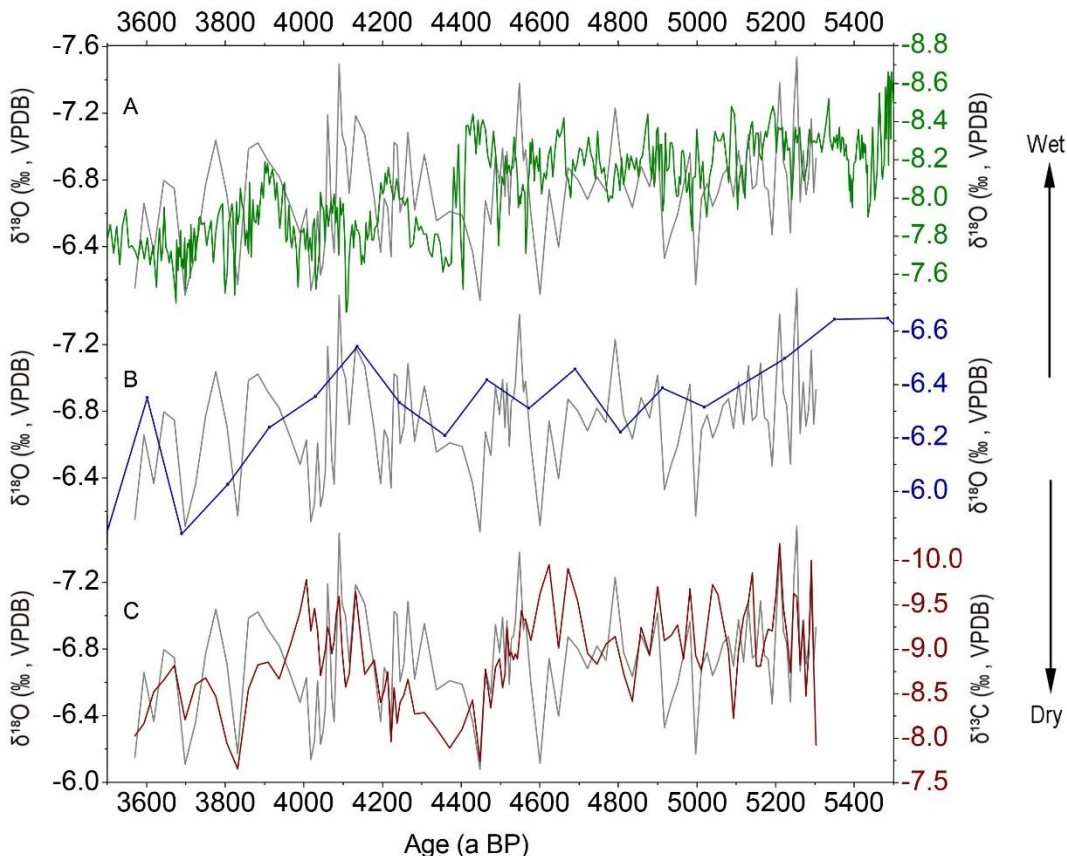

**Figure 6**. Replication of the δ¹⁸O records from three caves (Shennong Cave: grey lines, Dongge Cave: green line and Xiangshui Cave: dark blue line) during the overlapping growth period and comparison between δ¹⁸O (grey line) and δ¹³C (brown line) records of SN17 (C).

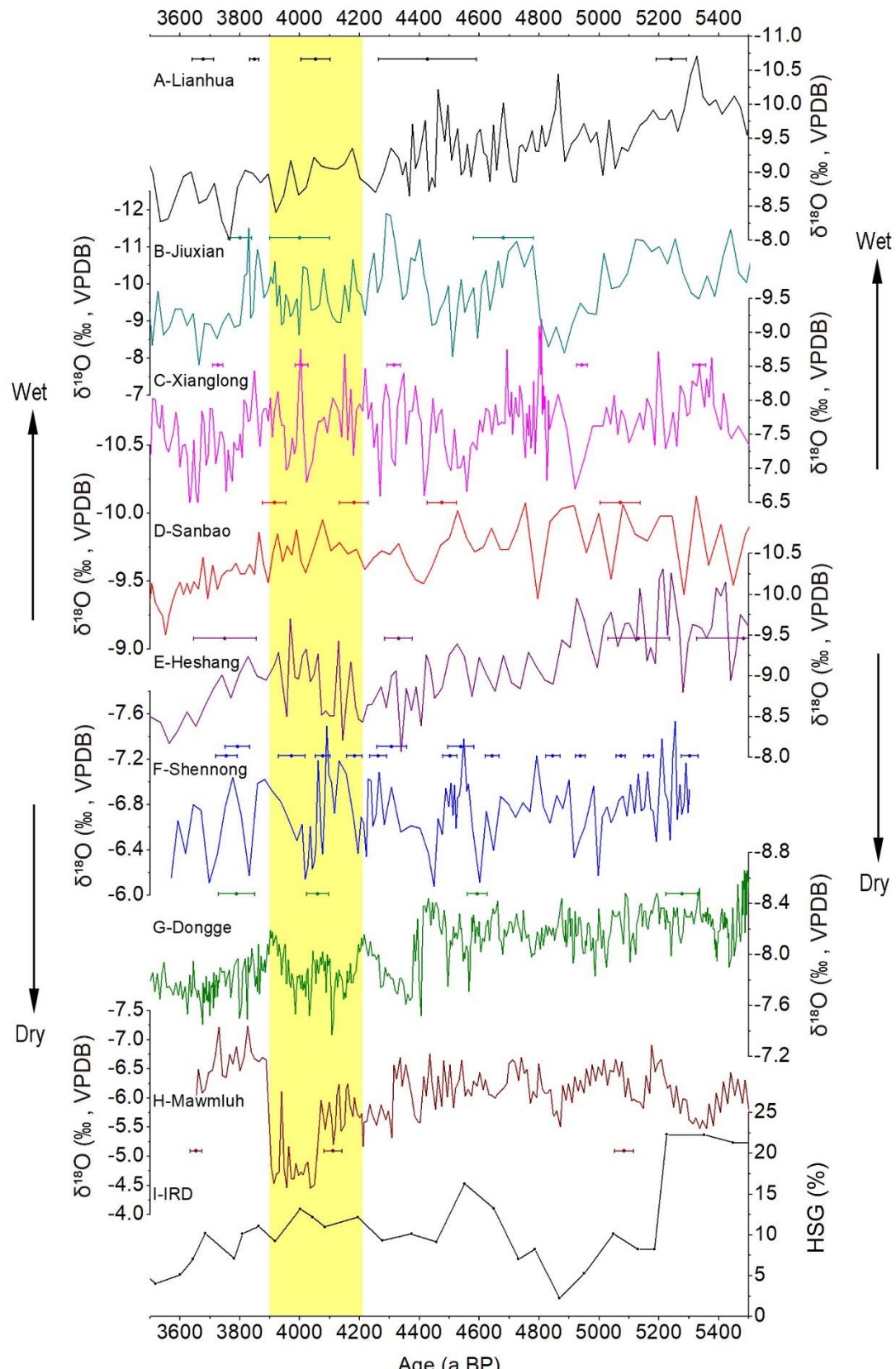

**Figure 7**. Comparison of δ¹⁸O records from (A) Lianhua Cave (Dong et al., 2015), (B) Jiuxian Cave (Cai et al., 2010), (C) Xianglong Cave (Tan et al., 2018a), (D) Sanbao Cave (Dong et al., 2010), (E) Heshang Cave (Hu et al., 2008), (F) Shennong Cave (this study), (G) Dongge Cave (Wang et al.,

2005), (H) Mawmluh Cave (Berkelhammer et al., 2012) and the ice-rafted hematite-stained grains (HSG) record from the North Atlantic (Bond et al., 2001). $^{230}$Th dates and error bars are illustrated with different colors for each stalagmite. Yellow bar marks the dry interval in northern and southwestern China and the wet interval in central and southeastern China during the 4.2 ka BP event (4.2-3.9 ka BP).

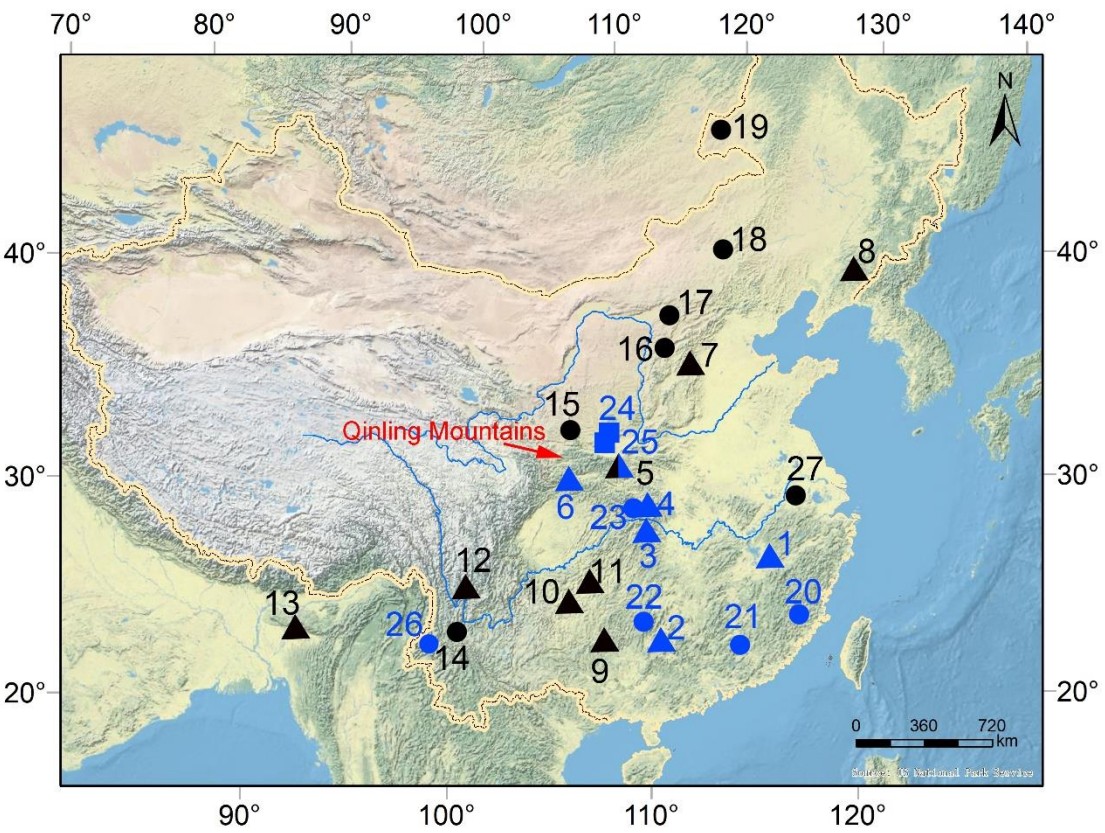

**Figure 8**. Map showing some locations discussed in the text (1-Shennong Cave (this study); 2-Xiangshui Cave (Zhang et al., 2004); 3-Heshang Cave (Hu et al., 2008); 4-Sanbao Cave (Dong et al., 2010); 5-Jiuxian Cave (Cai et al., 2010), 6-Xianglong Cave (Tan et al., 2018a); 7-Lianhua Cave (Dong et al., 2015); 8-Nuanhe Cave (Tan, 2005); 9-Dongge Cave (Wang et al., 2005); 10-Dark Cave (Jiang et al., 2013); 11-Shigao Cave (Jiang et al., 2012); 12-Xianren Cave (Zhang et al., 2006); 13-Mawmluh Cave (Berkelhammer et al., 2012); 14-Erhai Lake (Zhou et al., 2003); 15-Tianchi Lake (Zhao et al., 2010); 16-Gonghai Lake (Chen et al., 2015); 17-Daihai Lake (Xiao et al., 2018a); 18-Dali Lake (Xiao et al., 2008); 19-Hulun Lake (Xiao et al., 2018b); 20-Daiyunshan peat (Zhao et al., 2017), 21-Dahu peat (Zhou et al., 2004), 22-Daping peat (Zhong et al., 2010a); 23-Dajiuhu peat (Ma et al., 2008); 24-Chengjiachuan site (Huang et al., 2010); 25-Huxizhuang loess-soil profile (Huang et al., 2011); 26-Tengchongqinghai Lake (Zhang et al., 2017); 27-Gaochun profile (Yao et al., 2017); 28-Zhongqiao site (Wu et al., 2017)). Solid triangle, dot and square denote stalagmite records, lake sediment/peat records and paleoflood sediment records, respectively. Black and blue color indicate a dry and a wet climate during the 4.2 ka BP event, respectively. The base map is the same as that in figure 1.

**Table 1**. $^{230}$Th results of stalagmite SN17 from Shennong Cave. The errors are $2\sigma$.

| Distance from the top (mm) | Sample Number | $^{238}$U (ppb) | $^{232}$Th (ppt) | $^{230}$Th / $^{232}$Th (atomic x10$^{-6}$) | $d^{234}$U* (measured) | $^{230}$Th / $^{238}$U (activity) | $^{230}$Th Age (yr) (uncorrected) | $^{230}$Th Age (yr BP)*** (corrected) | $d^{234}$U$_{Initial}$** (corrected) |
|---|---|---|---|---|---|---|---|---|---|
| 3 | SN17-3 | 2063±1.9 | 1025±22 | 1388±33 | 235.5±1.4 | 0.0418±0.0004 | 3736±38 | 3668±38 | 238±1 |
| 7 | SN17-7 | 1146±1.6 | 2511±51 | 320±7 | 235.3±2.1 | 0.0426±0.0003 | 3820±24 | 3707±43 | 238±2 |
| 12 | SN17-12 | 1757±1.8 | 984±22 | 1302±33 | 233.6±1.5 | 0.0442±0.0005 | 3979±46 | 3897±47 | 236±2 |
| 17 | SN17-17 | 2617±5.7 | 132±10 | 14960±1115 | 243.2±2.4 | 0.0457±0.0003 | 4076±26 | 4007±26 | 246±2 |
| 32 | SN17-32 | 2535±4.9 | 195±11 | 10103±565 | 252.1±2.2 | 0.0472±0.0003 | 4187±27 | 4117±27 | 255±2 |
| 36 | SN17-36 | 3572±4.9 | 1814±39 | 1560±35 | 247.1±1.7 | 0.0481±0.0003 | 4279±29 | 4268±30 | 250±2 |
| 44 | SN17-44 | 1122±1.2 | 151±10 | 6034±420 | 265.5±1.9 | 0.0492±0.0006 | 4319±52 | 4248±52 | 269±2 |
| 51 | SN17-51 | 1035±1.7 | 1169±24 | 763±16 | 279.4±1.9 | 0.0522±0.0002 | 4540±18 | 4452±25 | 283±2 |
| 56 | SN17-56 | 2976±3.6 | 82±20 | 31024±7484 | 264.4±1.4 | 0.0519±0.0005 | 4560±18 | 4491±46 | 268±1 |
| 75 | SN17-75 | 4619±7.3 | 66±14 | 61740±12772 | 268.8±1.9 | 0.0532±0.0003 | 4668±25 | 4600±25 | 272±2 |
| 120 | SN17-120 | 4640±8.5 | 225±14 | 19262±1186 | 290.3±2.2 | 0.0566±0.0003 | 4883±24 | 4814±24 | 294±2 |
| 143 | SN17-143 | 2993±5.2 | 1031±21 | 2810±58 | 312.9±2.2 | 0.0587±0.0002 | 4980±17 | 4910±17 | 317±2 |
| 190 | SN17-190 | 6380±9.5 | 56±10 | 114548±20151 | 322.2±1.4 | 0.0608±0.0002 | 5120±16 | 5052±16 | 327±1 |
| 227 | SN17-227 | 4436±10.2 | 873±18 | 5189±106 | 323.7±2.2 | 0.0619±0.0002 | 5216±17 | 5149±17 | 329±2 |
| 317 | SN17-317 | 6886±31.0 | 2221±45 | 3273±67 | 331.7±3.0 | 0.0640±0.0003 | 5363±29 | 5294±29 | 337±3 |

U decay constants: $\lambda_{238}$ = 1.55125×10$^{-10}$ and $\lambda_{234}$ = 2.82206×10$^{-6}$. Th decay constant: $\lambda_{230}$ = 9.1705×10$^{-6}$ (ref. 56). *$\delta^{234}$U = ([$^{234}$U/$^{238}$U]$_{activity}$ − 1) ×1000. ** $\delta^{234}$U$_{initial}$ was calculated based on $^{230}$Th age (T), i.e., $\delta^{234}$U$_{initial}$ = $\delta^{234}$U$_{measured}$×e$^{\lambda234xT}$. Corrected $^{230}$Th ages assume the initial $^{230}$Th/$^{232}$Th atomic ratio of 4.4±2.2 ×10$^{-6}$. Those are the values for a material at secular equilibrium, with the bulk earth $^{232}$Th/$^{238}$U value of 3.8. The errors are arbitrarily assumed to be 50%.

755