# Peer review of "Hydroclimatic variations in southeastern China during the 4.2 ka event reflected by stalagmite records"

_Climate of the Past, 2018_

## Referee Comment (RC1) · Anonymous Referee #1 · 2 Oct 2018

General comments:

The heterogeneity of the 4.2 ka BP climatic event requires intensive researches of high-quality, high-resolution proxy records from climatically sensitive and geographically representative regions in order to reveal the spatiotemporal pattern of the event and the associated mechanism. This manuscript provided the East Asian summer monsoon with a new stalagmite record spanning the interval of 5.3-3.6 ka BP from a cave in southeast China where is a key gap of high-resolution climate records in the Asian monsoon region and investigated the possible north-south pattern of the monsoon precipitation during the 4.2 ka BP event based on the comparison of previously

published proxy records from southern and northern China. The data and inferences presented in the study are of great significance and would contribute to a better understanding of the mechanism responsible for East Asian summer monsoon variations on millennial to centennial scales. I recommend acceptance of this manuscript for publication in CP after revisions.

1. The manuscript interpreted the newly obtained stalagmite record and depicted the process of East Asian summer monsoon changes during the interval of 5.3-3.6 ka BP. Although it is necessary to do so, the 4.2 ka BP event itself should be paid more attention given that the manuscript is expected to contribute to the Special Issue "The 4.2 ka BP climatic event". I understand the authors' inference that the 4.2 ka BP event might manifest a wet spell in southern China but a dry spell in northern China. What is the timing of the 4.2 ka BP event occurring in monsoon China? When did it start and end in southern and northern China, respectively? Where does the boundary lies if the event displays different regional manifestations in northern and southern China? I suggest that the authors give more discussions about these issues.

2. The authors made a comparison between stalagmite records and peat ones to investigate the spatial manifestation of the 4.2 ka BP event in the monsoon region. As everyone knows, peat sequences are unparalleled in both dating precision and resolution with stalagmites. In view of the sufficient number of the published stalagmite records from the monsoon China, I suggest that the authors remove the peat records mentioned in the manuscript and focus on the existing stalagmites records.

Specific comments:

1. Abstract on lines 19-32. Better to clearly explain the nature, especially the timing of the 4.2 ka BP event in southern and northern China.

2. Lines 98-105. Is it possible to go over 1.5 km from the cave entrance to get stalagmites that consist of pure calcite? It is worthwhile if possible, because the cave lies in a key zone in monsoon China as shown in Figure 1.

3. Lines 117-119. State the purpose of sampling in different thickness intervals.

4. Lines 134-154. This paragraph, as a part of the results, should be focused on the description of features of $\delta18O$ and $\delta13C$ fluctuations on different timescales during the study interval. Remove the part regarding discussions of isotopic equilibrium (lines 140-151) to the next paragraph "4.1 Interpretation of $\delta18O$ and $\delta13C$".

5. Lines 158-169. Delete or reduce this part.

6. Lines 169-178. Show the location of E'mei cave in Figure 1, and add one Figure to show the correlation between the speleothem $\delta18O$ record from E'mei cave and the EASM precipitation amount in 1951-2009 AD.

7. Lines 179-192. Reduce this part and consider to integrate this part with the part on lines 140-151 to briefly explain 1) the relation between $\delta18O$ and $\delta13C$ (isotopic equilibrium), and 2) the implications of $\delta18O$ and $\delta13C$.

8. Lines 205-252. This paragraph should be organized only on the basis of the data obtained from this study. Remove lines 225-227 to "4.3". Delete lines 245-249. Remove lines 249-252 to "4.3". More importantly, rewrite a new paragraph on the basis of the part on lines 242-244 to explain the nature and timing of the 4.2 ka BP event reflected by the study stalagmite.

9. Lines 254-307. Delete the first paragraph on lines 255-271. The part "4.3" should give a clear view of 1) the nature and timing of the 4.2 ka BP event in southern and northern China, and the boundary between the dry north and the wet south based on the comparison of stalagmite records from monsoon China (eastern China).

10. Figure 1. Remove sites of the peat records from Panel A, and show SN in a different sign.

11. Figure 2. Remove "0" from the X axis of Panels A and B and show months consecutively (better as abbreviations in English). In Panel C, the tick marks for each time interval showing on the X axis seem to be one less.

12. Figure 7. Delete Panel E. Panel H should be $\delta$13C rather than $\delta$18O.

13. Table 1. Better to show the distance from the top for each sample.

Related aspects: 1. Does the paper address relevant scientific questions within the scope of CP? Yes. 2. Does the paper present novel concepts, ideas, tools, or data? Yes. 3. Are substantial conclusions reached? Yes. 4. Are the scientific methods and assumptions valid and clearly outlined? Yes. 5. Are the results sufficient to support the interpretations and conclusions? Yes. 6. Is the description of experiments and calculations sufficiently complete and precise to allow their reproduction by fellow scientists (traceability of results)? Yes. 7. Do the authors give proper credit to related work and clearly indicate their own new/original contribution Yes. 8. Does the title clearly reflect the contents of the paper? Yes. 9. Does the abstract provide a concise and complete summary? Not sufficient. 10. Is the overall presentation well structured and clear? Not sufficient. 11. Is the language fluent and precise? Yes. 12. Are mathematical formulae, symbols, abbreviations, and units correctly defined and used? Yes. 13. Should any parts of the paper (text, formulae, figures, tables) be clarified, reduced, combined, or eliminated? Yes. 14. Are the number and quality of references appropriate? Yes. 15. Is the amount and quality of supplementary material appropriate? Yes.

---

## Referee Comment (RC2) · Anonymous Referee #2 · 17 Oct 2018

General Comments: The manuscript describes a new, well-dated isotopic record of environmental change from the transitional period between the Middle and Late Holocene. Overall, the data appear high quality and collected/analyzed properly. The presentation of the material and interpretation of these data is generally good, but does require some additional thought and discussion. Some structure and figure design changes are needed, as indicated in specific and technical comments below. Generally, this review agrees with comments made by reviewer 1, although this reviewer believes the peat record to still be of importance as it offers evidence of replicated environmental change in a different proxy, despite chronological limitations. This reviewer believes that after restructuring (detailed well by reviewer 1) and addressing the con-

cerns below (particularly about isotopic interpretations of wet/dry), the manuscript will be acceptable for publication.

Specific Comments: 44: 'specific level' is unclear/vague. Do you mean a specific physical level (like a layer at a depth) or a geochemical threshold?

47: Be careful about assigning direct causality between the climate changes and societal responses. For some better studied sites, a direct impact of climate change leading to societal collapse may be well-established enough to confidently state such. However, for many others it may be more accurate to highlight that the climate and societal changes coincided and were likely associated, but not certainly proven. It is also important to acknowledge that climate-societal interactions are usually much more complex than our simplified paleo-perspectives (e.g., there was drought, so therefore their society suffered and collapsed).

76: Data source for Guixi data?

77: A sentence further explaining the climatic set up and characteristics of the spring persistent rainfall would help here, since it seems to be an interesting and important regional characteristic

82: "Data from..." sentence is convoluted and difficult to read as is.

87: Is it important that it was found after days of heavy rain? Was it previously not open/accessible?

92: You have some taxonomic inconsistency reporting plants here: Pinus is a genus, Taxodiaceae is a family. "Camelliaoleifera" should be a binomial genus/species: Camellia oleifera. Bamboo is only given as a common name. Preferably, you should list plants on the same taxonomic level (probably just genus), and species level is probably not necessary for your discussion here. Also, Taxodiaceae is no longer a recognized plant family; it has since been absorbed into Cupressaceae.

129: Was evidence of hiatus examined petrographically? Or is this conclusion simply

based on the age distribution? The top 50 mm have a few petrographic boundaries I can see in Fig 3 that might be worth examining closer for short hiatuses petrographically (if you haven't already done so) (e.g., Railsback 2013).

131: The linear age-depth model looks sufficient. It might be worthwhile to age model with BACON or StalAge and see if that changes any results/interpretation significantly.

150: It may be useful to note that even stalagmites that deposit with kinetic fractionation can still preserve valuable climate data BECAUSE of the fractionation. So even if your stalagmite isn't in isotopic equilibrium, it can still have useful data (though your interpretation of the isotopes may be different).

165: "orbital" is not a timescale. Millennial timescales should suffice. However, on the timescale you are examining, orbital forcings are not a factor, so this is a somewhat weak/irrelevant point. Focus on what the literature says about controls on d18O for the decadal/centennial range you are examining.

171: "We suggest"- Are you suggesting that conclusion newly in this paper? Or was this the conclusion of Zhang 2018 you cite? If the latter, I would rephrase to simply state that data from E'mei cave concluded that EASM-NSM balance controls the d18O, and not say "we" concluded it.

171: Earlier (65) you said there was only one published stalagmite from SE China, but isn't Zhang 2018 another published record from SE China?

178: A sentence clarifying and summarizing how you are interpreting the d18O in Shennong Cave would be nice here, since you state several possible ways to interpret d18O for the region.

190: Your d13C summary is generally good. Some supplemental resources you might want to examine include Oster et al., 2010; Meyer et al., 2014; Noronha et al., 2015;Wong and Breecker, 2015 to get more recent studies and summaries on d13C.

200: I think your dismissal of the effects of degassing and PCP is premature. Some

degassing must occur in order for CaCO3 precipitation to occur (being deposited in perfect 'isotopic equilibrium' is impossible, since a system in equilibrium will not undergo any reactions or change). And the presence of soda straws and stalactites (which I assume are present in the cave) means PCP is also occurring. The negative relationship between d13C and growth rate suggest to me that PCP is perhaps quite important as a control. Perhaps more importantly, you could argue that vegetation dynamics are a major or the major control on d13C, but when multiple factors are working in concert (e.g., drier conditions both lead to less vegetation and greater PCP which both lead to higher d13C values), dismissing one or more potential factors is not even necessary.

214: Do you have any supporting evidence that the d18O for your stalagmite reflects annual precip (e.g., through drip water monitoring?) or is this an assumption? I think the match between it and Dongge make a decent argument that your stal is recording long-term aggregates rather than 'flashy' storm events. But how you decided that it is annual precip should be mentioned.

222: Wouldn't wet intervals be those with z scores less than zero? (Not greater, like you have written). Also, You earlier state that d18O is interpreted in your area as the ratio between EASM and NSM amounts, with lower values meaning a greater fraction of EASM. Shifting the seasonality of precipitation can therefore change the d18O in the stalagmite without actually changing annual precipitation amounts. Additionally, a stalagmite d18O that decreases could be because the EASM gets more intense (more overall rainfall), but also when the NSM decreases more than the EASM (less overall rainfall). Be careful about interpreting d18O as amount unless you have supporting evidence.

223: "More wet intervals": Wouldn't a better measurement be "more years wetter than average"? This sounds like you are just counting the number of times you have a span below 0 Z-score (so a highly variable record with many changes above and below average could easily have more 'wet intervals' than a record that is all wetter than average in a single long 'wet interval').

[Figure]

227: Controlled by what variable of summer monsoon precip? Amount? EASM/NSM like your cave?

230: Growth rate is often not a direct function of precipitation amount (e.g., Railsback 2018). If you believe growth rate in your stal is a direct relationship to precip amount, some supporting evidence/arguments would be beneficial.

238: 150 years is a pretty long time to be a vegetational response delay in terms of vegetation coverage, particularly if there is not a significant shift in vegetation type. Do you have a more detailed explanation of why the vegetational response would take 150 years? Are there alternative reasons that could explain the lag? Perhaps d13C is showing actual precip amount changes, and the 'lag' is because the d18O can reflect proportional shifts in EASM/NSM that may not result in actual precip amount changes.

255:The previous paragraph contained records in monsoonal China covering the 4.2 ka event. Why are they separate from section 4.3?

287: While the argument linking EASM intensity to AMOC is sound, the IRD record is not particularly strong evidence since the variance in IRD between 3700 and 4500 yr BP is quite small. Are there alternative records for AMOC intensity you could use, or perhaps support this by bringing in records also showing monsoonal changes in Africa and South Asia at this time.

294: Are you calculating coherence, or do you mean the variables co-vary?

Fig 1: Map A is too far zoomed out and is difficult to see sites. Ideally would have main part of map and this figure focused on eastern China. Small inset map could provide wider context. Maps also need a legend identifying icons and color scheme for basemap. Highlight your site on main map better (e.g., larger text, unique color, pointing arrow). Another map or layer on this map showing typical modern location of the summer monsoon influence/extent would be beneficial. No scale on map B.

610: You bring up several more climatic influencing winds here that are never discussed

[Figure]

or mentioned in your paper. If they are important, they need to be discussed, or at least mentioned why you are not considering them.

Fig 4: Labeling the y axes with the environmental interpretation (e.g., wetter/drier, more intense EASM, etc) would aid the understanding of these plots

Fig 6: Labeling the axes with the cave name (or directly on the plot) along with in the caption would make the plot more readable. The coarse resolution of Xiangshui makes it very difficult for me to conclude anything about the covariation between it and your record. I do think the Dongge records visually matches well.

Fig 7: Labeling the axes with the sample/cave/site name (or directly on the plot) along with in the caption would make the plot more readable. Also, labeling the Y-axes with the environmental interpretation (wet/dry, monsoon N-S offset, etc) will help.

Fig 7: The yellow bars don't align well with your d18O record. Is there a reason they are offset from the low value intervals of your record?

Technical comments: 73: Your latitude/longitude is flipped

81: Shennong Cave or Shennong cave? Capitalization consistency. Also, this sentence seems unnecessary and out of place as you already mentioned that the cave is in the region of spring persistent rainfall.

160: Re-examine your use of commas in this sentence. It's unclear which phrases are meant to be grouped in the list of influences.

188: Prior not needed to be capitalized

196: Cave or cave? I think that Cave should be used when referring to specific named caves here and throughout, but it's more important for you to be consistent with capitalization.

211: 'wetter to drier conditions' is better, because there wasn't a major regime shift into definitively 'dry' conditions from earlier 'wet' conditions

255: "A remarkable drought" is better here than "the remarkable drought", since you haven't discussed the drought for the past few pages.

268: "the large dating uncertainties and the low resolution" Change to "by large dating uncertainties and low proxy resolution in many records"? or something more clear

610: Westerly used here sounds like you are saying westerly monsoon, but you are probably just referring to the westerlies, correct?

Figure 2: You may wish to recolor the portion with the red-green lines. Almost 1 in 10 people suffer from some degree of red-green colorblindness.

Fig 3: Just a design though: Your age markers are red on the plot, but black on the stalagmite. The red marks on the stalagmite are XRD. For consistency and ease of eye-matching of this figure, you might consider making the age markers on the stalagmite red and the XRD markers a different color.

Figure 4: You may wish to recolor the portion with the contrasting red-green Z score. Almost 1 in 10 people suffer from some degree of red-green colorblindness.

---

## Author Comment (AC1) · 24 Oct 2018

Response to referee #1.

General comments: The heterogeneity of the 4.2 ka BP climatic event requires intensive researches of high-quality, high-resolution proxy records from climatically sensitive and geographically representative regions in order to reveal the spatiotemporal pattern of the event and the associated mechanism. This manuscript provided the East Asian summer monsoon with a new stalagmite record spanning the interval of 5.3-3.6 ka BP from a cave in southeast China where is a key gap of high-resolution climate records in the Asian monsoon region and investigated the possible north-south pattern of the

monsoon precipitation during the 4.2 ka BP event based on the comparison of previously published proxy records from southern and northern China. The data and inferences presented in the study are of great significance and would contribute to a better understanding of the mechanism responsible for East Asian summer monsoon variations on millennial to centennial scales. I recommend acceptance of this manuscript for publication in CP after revisions.

1. The manuscript interpreted the newly obtained stalagmite record and depicted the process of East Asian summer monsoon changes during the interval of 5.3-3.6 ka BP. Although it is necessary to do so, the 4.2 ka BP event itself should be paid more attention given that the manuscript is expected to contribute to the Special Issue "The 4.2 ka BP climatic event". I understand the authors' inference that the 4.2 ka BP event might manifest a wet spell in southern China but a dry spell in northern China. What is the timing of the 4.2 ka BP event occurring in monsoon China? When did it start and end in southern and northern China, respectively? Where does the boundary lies if the event displays different regional manifestations in northern and southern China? I suggest that the authors give more discussions about these issues. Answer# Thanks very much for your suggestions. We agree with you. Firstly, we will focus on the discussion of the timing and structure of the 4.2 ka BP event. Secondly, we will discuss the start and end timing of the 4.2 ka BP event in northern and southern China, respectively. Thirdly, we will give the boundary between the dry north and the wet south.

2. The authors made a comparison between stalagmite records and peat ones to investigate the spatial manifestation of the 4.2 ka BP event in the monsoon region. As everyone knows, peat sequences are unparalleled in both dating precision and resolution with stalagmites. In view of the sufficient number of the published stalagmite records from the monsoon China, I suggest that the authors remove the peat records mentioned in the manuscript and focus on the existing stalagmites records. Answer# Agree. The peat records will be removed from the comparison (figure 7), however, we

will still mention some of them in the introduction and use them as possible evidences in the discussion.

Specific comments: 1. Abstract on lines 19-32. Better to clearly explain the nature, especially the timing of the 4.2 ka BP event in southern and northern China. Answer# Agree, it will be revised.

2. Lines 98-105. Is it possible to go over 1.5 km from the cave entrance to get stalagmites that consist of pure calcite? It is worthwhile if possible, because the cave lies in a key zone in monsoon China as shown in Figure 1. Answer# Yes, the calcite speleothems can be found in the more distal parts of the cave. We have already described this in lines 99-101. This was also discussed in another published paper by Zhang et al. (2015), which will be cited in revision.

3. Lines 117-119. State the purpose of sampling in different thickness intervals. Answer# 0-75 mm from the top was deposited between 3.7 and 4.6 ka BP with low growth rate, higher resolution obtained by increasing sampling interval in this section could show more detailed information around 4.2 ka BP. It will be clearly stated.

4. Lines 134-154. This paragraph, as a part of the results, should be focused on the description of features of _18O and _13C fluctuations on different timescales during the study interval. Remove the part regarding discussions of isotopic equilibrium (lines 140-151) to the next paragraph "4.1 Interpretation of _18O and _13C". Answer# Agree, the features of $\delta$18O and $\delta$13C records will be described in detailed and lines 140-151 will be removed to section 4.1.

5. Lines 158-169. Delete or reduce this part. Answer# It will be reduced.

6. Lines 169-178. Show the location of E'mei cave in Figure 1, and add one Figure to show the correlation between the speleothem _18O record from E'mei cave and the EASM precipitation amount in 1951-2009 AD. Answer# We will show the location of E'mei cave in Figure 1 and add one figure about the correlation between E'mei $\delta$18O

record and the precipitation amount in 1951-2009AD.

7. Lines 179-192. Reduce this part and consider to integrate this part with the part on lines 140-151 to briefly explain 1) the relation between _18O and _13C (isotopic equilibrium), and 2) the implications of _18O and _13C. Answer# Agree, it will be revised.

8. Lines 205-252. This paragraph should be organized only on the basis of the data obtained from this study. Remove lines 225-227 to "4.3". Delete lines 245-249. Remove lines 249-252 to "4.3". More importantly, rewrite a new paragraph on the basis of the part on lines 242-244 to explain the nature and timing of the 4.2 ka BP event reflected by the study stalagmite. Answer# Agree, we will revise this paragraph according to your suggestions.

9. Lines 254-307. Delete the first paragraph on lines 255-271. The part "4.3" should give a clear view of 1) the nature and timing of the 4.2 ka BP event in southern and northern China, and the boundary between the dry north and the wet south based on the comparison of stalagmite records from monsoon China (eastern China). Answer# Agree, we will discuss the nature and timing of the 4.2 ka BP event in southern and northern China, and the boundary between the dry north and the wet south.

10. Figure 1. Remove sites of the peat records from Panel A, and show SN in a different sign. Answer# Agree, we will redraw figure 1.

11. Figure 2. Remove "0" from the X axis of Panels A and B and show months consecutively (better as abbreviations in English). In Panel C, the tick marks for each time interval showing on the X axis seem to be one less. Answer# Agree, we will redraw figure 2.

12. Figure 7. Delete Panel E. Panel H should be _13C rather than _18O. Answer# Agree, we will redraw it according to your suggestion.

13. Table 1. Better to show the distance from the top for each sample. Answer# Agree, we will add one column showing the distance from the top for each sample.

Related aspects: 1. Does the paper address relevant scientific questions within the scope of CP? Yes. 2. Does the paper present novel concepts, ideas, tools, or data? Yes. 3. Are substantial conclusions reached? Yes. 4. Are the scientific methods and assumptions valid and clearly outlined? Yes. 5. Are the results sufficient to support the interpretations and conclusions? Yes. 6. Is the description of experiments and calculations sufficiently complete and precise to allow their reproduction by fellow scientists (traceability of results)? Yes. 7. Do the authors give proper credit to related work and clearly indicate their own new/original contribution Yes. 8. Does the title clearly reflect the contents of the paper? Yes. 9. Does the abstract provide a concise and complete summary? Not sufficient. 10. Is the overall presentation well structured and clear? Not sufficient. 11. Is the language fluent and precise? Yes. 12. Are mathematical formulae, symbols, abbreviations, and units correctly defined and used? Yes. 13. Should any parts of the paper (text, formulae, figures, tables) be clarified, reduced, combined, or eliminated? Yes. 14. Are the number and quality of references appropriate? Yes. 15. Is the amount and quality of supplementary material appropriate? Yes. Answer# Thanks very much for your critical comments and helpfull suggestions. We will revise the manuscript according to your suggestions.

---

## Author Comment (AC2) · 24 Oct 2018

Response to referee #2

General Comments: The manuscript describes a new, well-dated isotopic record of environmental change from the transitional period between the Middle and Late Holocene. Overall, the data appear high quality and collected/analyzed properly. The presentation of the material and interpretation of these data is generally good, but does require some additional thought and discussion. Some structure and figure design changes are needed, as indicated in specific and technical comments below. Generally, this review agrees with comments made by reviewer 1, although this reviewer

believes the peat record to still be of importance as it offers evidence of replicated environmental change in a different proxy, despite chronological limitations. This reviewer believes that after restructuring (detailed well by reviewer 1) and addressing the concerns below (particularly about isotopic interpretations of wet/dry), the manuscript will be acceptable for publication.

Specific Comments: 44: 'specific level' is unclear/vague. Do you mean a specific physical level (like a layer at a depth) or a geochemical threshold? Answer# We will explain it as "the mineral of that stalagmite show a transformation from calcite to aragonite with -2‰ abrupt increasing of $\delta$18O values......"

47: Be careful about assigning direct causality between the climate changes and societal responses. For some better studied sites, a direct impact of climate change leading to societal collapse may be well-established enough to confidently state such. However, for many others it may be more accurate to highlight that the climate and societal changes coincided and were likely associated, but not certainly proven. It is also important to acknowledge that climate-societal interactions are usually much more complex than our simplified paleo-perspectives (e.g., there was drought, so therefore their society suffered and collapsed). Answer# Agree, we will change this expression as "the abrupt climate change associated with the 4.2 ka BP event was considered as a possible cause for the collapses of Neolithic cultures in China......"

76: Data source for Guixi data? Answer# Yes, it will be clearly stated.

77: A sentence further explaining the climatic set up and characteristics of the spring persistent rainfall would help here, since it seems to be an interesting and important regional characteristic 82: "Data from..." sentence is convoluted and difficult to read as is. Answer# A sentence will be added to explain the climatic characteristics in the region of spring persistent rain. The sentence "Data from..." will be rewritten. Such as "in the region of spring persistent rain, the EASM (May to September) precipitation accounts from accounts for 54% of the annual precipitation and the non-summer

monsoon (NSM, October to next April) precipitation accounts for 46%. Data from the nearest GNIP station in Changsha, also located in the region of the spring persistent rain, indicate that the $\delta$18O values of EASM precipitation are lower comparing with that of NSM precipitation. Therefore, different from the speleothem $\delta$18O from southwestern and northern part of the monsoonal China are mainly influenced by EASM precipitation, speleothem $\delta$18O in the spring persistent ring area are controlled by both EASM and NSM precipitation (Zhang et al., 2018)……."

87: Is it important that it was found after days of heavy rain? Was it previously not open/accessible? Answer# No, we just described how this cave was found. Before the heavy rain in 1998, nobody knows this cave. This sentence will be deleted.

92: You have some taxonomic inconsistency reporting plants here: Pinus is a genus, Taxodiaceae is a family. "Camelliaoleifera" should be a binomial genus/species: Camellia oleifera. Bamboo is only given as a common name. Preferably, you should list plants on the same taxonomic level (probably just genus), and species level is probably not necessary for your discussion here. Also, Taxodiaceae is no longer a recognized plant family; it has since been absorbed into Cupressaceae. Answer# Agree. It will be changed to "the overlying vegetation consists mainly of secondary forest tree species such as Pinus, Cunninghamia and Phyllostachys and shrub-like Camellia oleifera and Ilex which are C3 plants (Zhang et al., 2015)."

129: Was evidence of hiatus examined petrographically? Or is this conclusion simply based on the age distribution? The top 50 mm have a few petrographic boundaries I can see in Fig 3 that might be worth examining closer for short hiatuses petrographically (if you haven't already done so) (e.g., Railsback 2013). Answer# We did not check the petrography, it might be two short hiatuses at 10 and 50 mm. Thin section cannot be done in our lab, we think we might not have enough time to do this. We will increase more U-Th ages above and below these two layers to check the hiatuses, this is easy for our lab.

131: The linear age-depth model looks sufficient. It might be worthwhile to age model with BACON or StalAge and see if that changes any results/interpretation significantly. Answer# We will use COPRA age model.

150: It may be useful to note that even stalagmites that deposit with kinetic fractionation can still preserve valuable climate data BECAUSE of the fractionation. So even if your stalagmite isn't in isotopic equilibrium, it can still have useful data (though your interpretation of the isotopes may be different). Answer# Thanks for your suggestion. Currently, the replication of the $\delta$18O records between Shennong and Dongge Caves confirms that stalagmite SN17 was most likely deposited close to isotopic equilibrium. Following "Hendy test", we will analyze twenty-one subsamples from three growth layers to check whether it was deposited at or close to isotopic equilibrium. If not, we will note what you suggested.

165: "orbital" is not a timescale. Millennial timescales should suffice. However, on the timescale you are examining, orbital forcings are not a factor, so this is a somewhat weak/irrelevant point. Focus on what the literature says about controls on d18O for the decadal/centennial range you are examining. Answer# Agree, it will be revised.

171: "We suggest"- Are you suggesting that conclusion newly in this paper? Or was this the conclusion of Zhang 2018 you cite? If the latter, I would rephrase to simply state that data from E'mei cave concluded that EASM-NSM balance controls the d18O, and not say "we" concluded it. Answer# Agree, it will be corrected.

171: Earlier (65) you said there was only one published stalagmite from SE China, but isn't Zhang 2018 another published record from SE China? Answer# No, it's another paper (Zhang et al., 2004), a record from Xiangshui Cave. It will be clearly stated and cited in the revision. Zhang, M., Yuan, D. X., Lin, Y., Qin, J., Bin, L., Cheng, H., and Edwards, R. L.: A 6000-year high-resolution climatic record from a stalagmite in Xiangshui Cave, Guilin, China, The Holocene, 14, 697, 2004.

178: A sentence clarifying and summarizing how you are interpreting the d18O in

Shennong Cave would be nice here, since you state several possible ways to interpret d18O for the region. Answer# Agree. We will add one sentence.

190: Your d13C summary is generally good. Some supplemental resources you might want to examine include Oster et al., 2010; Meyer et al., 2014; Noronha et al., 2015; Wong and Breecker, 2015 to get more recent studies and summaries on d13C. Answer# Thanks for your recommendation, we will read and cite them.

200: I think your dismissal of the effects of degassing and PCP is premature. Some degassing must occur in order for CaCO3 precipitation to occur (being deposited in perfect 'isotopic equilibrium' is impossible, since a system in equilibrium will not undergo any reactions or change). And the presence of soda straws and stalactites (which I assume are present in the cave) means PCP is also occurring. The negative relationship between d13C and growth rate suggest to me that PCP is perhaps quite important as a control. Perhaps more importantly, you could argue that vegetation dynamics are a major or the major control on d13C, but when multiple factors are working in concert (e.g., drier conditions both lead to less vegetation and greater PCP which both lead to higher d13C values), dismissing one or more potential factors is not even necessary. Answer# Agree. We will revise this part.

214: Do you have any supporting evidence that the d18O for your stalagmite reflects annual precip (e.g., through drip water monitoring?) or is this an assumption? I think the match between it and Dongge make a decent argument that your stal is recording long-term aggregates rather than 'flashy' storm events. But how you decided that it is annual precip should be mentioned. Answer# Yes, we have done monitoring work in this cave for two years, seasonal variation of drip water $\delta$18O shows very small variation ($\sim$6% in the whole year), consistent will amount-weighted annual precipitation $\delta$18O outside the cave. We will clearly state this in the revision.

222: Wouldn't wet intervals be those with z scores less than zero? (Not greater, like you have written). Also, You earlier state that d18O is interpreted in your area as the

ratio between EASM and NSM amounts, with lower values meaning a greater fraction of EASM. Shifting the seasonality of precipitation can therefore change the d18O in the stalagmite without actually changing annual precipitation amounts. Additionally, a stalagmite d18O that decreases could be because the EASM gets more intense (more overall rainfall), but also when the NSM decreases more than the EASM (less overall rainfall). Be careful about interpreting d18O as amount unless you have supporting evidence. Answer# In our published paper (Zhang et al., 2018), we found that E'mei $\delta$18O record is significantly negatively correlated with the ratio of EASM/NSM precipitation (r=-0.67, p< 0.01) and the EASM precipitation (r= -0.54, p< 0.01). In addition, E'mei $\delta$18O record exhibits a coherent variation with the drought/flood index (reconstructed by historical records in summer) during 1810-2010 AD on decadal to centennial timescales. It indicates that E'mei $\delta$18O record can be also influenced by EASM precipitation amount (not annual precipitation amount) on decadal to centennial timescales, although EASM/NSM can better explain it on interannual timescales. We will clearly state this in the revision.

223: "More wet intervals": Wouldn't a better measurement be "more years wetter than average"? This sounds like you are just counting the number of times you have a span below 0 Z-score (so a highly variable record with many changes above and below average could easily have more 'wet intervals' than a record that is all wetter than average in a single long 'wet interval'). Answer# We will revise it.

227: Controlled by what variable of summer monsoon precip? Amount? EASM/NSM like your cave? Answer# It's summer monsoon (i.e., EASM) precipitation amount. We will clearly state it.

230: Growth rate is often not a direct function of precipitation amount (e.g., Railsback 2018). If you believe growth rate in your stal is a direct relationship to precip amount, some supporting evidence/arguments would be beneficial. Answer# We will revise it.

238: 150 years is a pretty long time to be a vegetational response delay in terms of

vegetation coverage, particularly if there is not a significant shift in vegetation type. Do you have a more detailed explanation of why the vegetational response would take 150 years? Are there alternative reasons that could explain the lag? Perhaps d13C is showing actual precip amount changes, and the 'lag' is because the d18O can reflect proportional shifts in EASM/NSM that may not result in actual precip amount changes. Answer# Agree. We will revise this part.

255:The previous paragraph contained records in monsoonal China covering the 4.2 ka event. Why are they separate from section 4.3? Answer# We will combine them together.

287: While the argument linking EASM intensity to AMOC is sound, the IRD record is not particularly strong evidence since the variance in IRD between 3700 and 4500 yr BP is quite small. Are there alternative records for AMOC intensity you could use, or perhaps support this by bringing in records also showing monsoonal changes in Africa and South Asia at this time. Answer# Thanks for your suggestion. A record from India will be added for comparison and discussion.

294: Are you calculating coherence, or do you mean the variables co-vary? Answer# We mean co-vary, we will also calculate the coherence.

Fig 1: Map A is too far zoomed out and is difficult to see sites. Ideally would have main part of map and this figure focused on eastern China. Small inset map could provide wider context. Maps also need a legend identifying icons and color scheme for basemap. Highlight your site on main map better (e.g., larger text, unique color, pointing arrow). Another map or layer on this map showing typical modern location of the summer monsoon influence/extent would be beneficial. No scale on map B. 610: You bring up several more climatic influencing winds here that are never discussed or mentioned in your paper. If they are important, they need to be discussed, or at least mentioned why you are not considering them. Fig 4: Labeling the y axes with the environmental interpretation (e.g., wetter/drier, more intense EASM, etc) would aid the

understanding of these plots Fig 6: Labeling the axes with the cave name (or directly on the plot) along with in the caption would make the plot more readable. The coarse resolution of Xiangshui makes it very difficult for me to conclude anything about the covariation between it and your record. I do think the Dongge records visually matches well. Fig 7: Labeling the axes with the sample/cave/site name (or directly on the plot) along with in the caption would make the plot more readable. Also, labeling the Y-axes with the environmental interpretation (wet/dry, monsoon N-S offset, etc) will help. Fig 7: The yellow bars don't align well with your d18O record. Is there a reason they are offset from the low value intervals of your record?

Answer# Agree. We will redraw all of these figures according to your suggestions.

Technical comments: 73: Your latitude/longitude is flipped Answer# Agree. It will be corrected.

81: Shennong Cave or Shennong cave? Capitalization consistency. Also, this sentence seems unnecessary and out of place as you already mentioned that the cave is in the region of spring persistent rainfall. Answer# Agree. It will be corrected.

160: Re-examine your use of commas in this sentence. It's unclear which phrases are meant to be grouped in the list of influences. Answer# We will rewrite this sentence.

188: Prior not needed to be capitalized Answer# It will be corrected.

196: Cave or cave? I think that Cave should be used when referring to specific named caves here and throughout, but it's more important for you to be consistent with capitalization. Answer# It will be corrected.

211: 'wetter to drier conditions' is better, because there wasn't a major regime shift into definitively 'dry' conditions from earlier 'wet' conditions Answer# It will be corrected.

255: "A remarkable drought" is better here than "the remarkable drought", since you haven't discussed the drought for the past few pages. Answer# It will be corrected.

268: "the large dating uncertainties and the low resolution" Change to "by large dating uncertainties and low proxy resolution in many records"? or something more clear Answer# It will be corrected.

610: Westerly used here sounds like you are saying westerly monsoon, but you are probably just referring to the westerlies, correct? Answer# It will be corrected.

Figure 2: You may wish to recolor the portion with the red-green lines. Almost 1 in 10 people suffer from some degree of red-green colorblindness. Answer# It will be redrawn.

Fig 3: Just a design though: Your age markers are red on the plot, but black on the stalagmite. The red marks on the stalagmite are XRD. For consistency and ease of eyematching of this figure, you might consider making the age markers on the stalagmite red and the XRD markers a different color. Answer# It will be redrawn.

Figure 4: You may wish to recolor the portion with the contrasting red-green Z score. Almost 1 in 10 people suffer from some degree of red-green colorblindness. Answer# It will be redrawn.

---

## Short Comment (SC1) · 25 Oct 2018

I am a Master's student with an interest in the research front.

Summary.

The purpose of this paper is to examine the link between environmental changes in the 5.3 and 3.6 ka BP period and the collapse of several Neolithic cultures in China like the Shijiahe Culture which was located in the middle reaches of the Yangtze River. Bigger focus is given on the 4.2 ka BP event during which dryer conditions insisted on the northern parts of China compared to the southern part, where the conditions where

more wet. This is done by reconstructing regional monsoon intensity from a stalagmite in Shennong Cave (SN17). The reconstruction is done through $\delta$18O and $\delta$13C proxy analysis. The results are cross-referenced with other proxy data from previous work done on China monsoon. The explanation claimed for these climatic changes on the 4.2 ka BP event are attributed to a weaker East Asian summer monsoon (EASM) due to reduced Atlantic Meridional Overturning Circulation (AMOC) which led to a southward migration of the Intertropical Convergence Zone.

Paleoenvironmental reconstruction research to help us understand about the conditions surrounding the development of past civilizations and cultures is very important. This paper can give another aspect on this research front, with a greater focus on Southeastern Asia.

Comments.

(1). The methods chapter was very simply put with references provided so if readers are further interested they could further examine the details of techniques used.

(2). This paper's results can give further information on how the EASM intensity can be derived from $\delta$18O concentration in speleothems.

(3). Are there enough evidence which support that ice-rafted debris decreased AMOC intensity during that period?

(4). E'mei cave is being mentioned on 171 line without the being labeled on Figure 1 where all the other sites are recorded. Since the regional environmental changes have a spatial significance it would be appropriate if it was presented.

(5). Various parameters are presented that are able to influence the $\delta$18O of the paper's speleothem. It would be useful that the conditions of these parameters were presented for the other speleothems used in this paper to compare with the data acquired by the authors.

Also to this note it may be good if the paper urges other researches into further investigating the reasons behind the amplitude difference between the SN17 and Jiuxian and Xianglong speleothems.

(6). Shennong Cave is located in an area that is influenced also from spring persistent rain. Maybe this is something that needs further investigation since it provides the area with a surplus of water compared to the more naturally dry northern part of China. Do the authors think that this is something worth considering?

(7). In Figure 6 there is a good correlation in the $\delta$18O records between the SN17, Dongee and Xianshui during the 4.2 ka BP event, but for the other dates the fluctuations vary considerably. How can one examine the effect that different spatial distribution of precipitation could have on this environment.

(8). In Figure 7.H the $\delta$13C values for SN17 should be displayed according to the header underneath. The scale seems to be wrong and should be corrected.

(9). In Figure 7 $\delta$18O concentration on C speleothem is lesser indicating intensified monsoon at the 4.2 ka BP event. In contrast to A,C, D where the concentration has higher values. C and D are very near and in a slightly lower latitude whereas A is slightly westwards. What would be the explanation for these values given the spatial connection of those speleothems?

---

## Short Comment (SC2) · 3 Nov 2018

Short comment by C. Tziavaras

The purpose of this paper is to examine the link between environmental changes in the 5.3 and 3.6 ka BP period and the collapse of several Neolithic cultures in China like the Shijiahe Culture which was located in the middle reaches of the Yangtze River. Bigger focus is given on the 4.2 ka BP event during which dryer conditions insisted on the northern parts of China compared to the southern part, where the conditions where more wet. This is done by reconstructing regional monsoon intensity from a stalagmite in Shennong Cave (SN17). The reconstruction is done through _18O and _13C proxy

analysis. The results are cross-referenced with other proxy data from previous work done on China monsoon. The explanation claimed for these climatic changes on the 4.2 ka BP event are attributed to a weaker East Asian summer monsoon (EASM) due to reduced Atlantic Meridional Overturning Circulation (AMOC) which led to a southward migration of the Intertropical Convergence Zone. Paleoenvironmental reconstruction research to help us understand about the conditions surrounding the development of past civilizations and cultures is very important. This paper can give another aspect on this research front, with a greater focus on Southeastern Asia.

Thanks very much for your comments and suggestions.

Comments. (1). The methods chapter was very simply put with references provided so if readers are further interested they could further examine the details of techniques used.

Answer: Yes, the related references were provided. Because the methods of U-Th dating and stable isotope measurement are extensively used in speleothem and paleoclimatic studies, the readers can find the similar descriptions in many papers.

(2). This paper's results can give further information on how the EASM intensity can be derived from _18O concentration in speleothems.

Answer: We discussed the significance of the speleothem $\delta$18O from the region of spring persistent rain in southeastern China. According to two reviewers' suggestions, we will also add more discussion about this issue in revision.

(3). Are there enough evidence which support that ice-rafted debris decreased AMOC intensity during that period?

Answer: Not enough, AMOC might be a possible reason. In the published papers, such as Wang et al. (2001), Tan et al. (2011, 2018), Chiang et al. (2015) and Railsback et al. (2018), a reduced AMOC was considered a possible reason causing a southward migration of ITCZ and a weakened East Asian summer monsoon. The weakened

monsoon during 4.2 ka BP event corresponds to higher amounts of ice-draft debris in North Atlantic (Figure 7I). The CPD paper Yan et al. (2018), published in the special issue 4.2 ka BP event, discussed the possible reason of 4.2 ka BP event using a set of long-term climate simulations. In their paper, all-forcing experiment show that the 4.2 ka BP event could be related to the slowdown of the AMOC, and the comparison between the all-forcing experiment and the single-forcing experiments indicates that the event was likely caused by internal variability. We will discuss these in the revision.

(4). E'mei cave is being mentioned on 171 line without the being labeled on Figure 1 where all the other sites are recorded. Since the regional environmental changes have a spatial significance it would be appropriate if it was presented.

Answer: The location of E'mei Cave will be shown on Figure 1.

(5). Various parameters are presented that are able to influence the _18O of the paper's speleothem. It would be useful that the conditions of these parameters were presented for the other speleothems used in this paper to compare with the data acquired by the authors. Also to this note it may be good if the paper urges other researches into further investigating the reasons behind the amplitude difference between the SN17 and Jiuxian and Xianglong speleothems.

Answer: According to the second reviewer's suggestion and yours, we will add more discussions about the significance of speleothem $\delta$18O from Shennong Cave in the revision. In addition, we will discuss the timing and nature of 4.2 ka BP event in northern and southern China by comparing with different speleothem $\delta$18O records from monsoonal China. Therefore, we will also discuss the reasons behind the amplitude difference among these records, which might be related to the rainbelt shift derived from the intensity variations in East Asian summer monsoon.

(6). Shennong Cave is located in an area that is influenced also from spring persistent rain. Maybe this is something that needs further investigation since it provides the area with a surplus of water compared to the more naturally dry northern part of China. Do

the authors think that this is something worth considering?

Answer: We will describe more details about the seasonal variabilities of precipitation amount and $\delta18O$ in the region of spring persistent rain, which is different from the other regions in the monsoonal China.

(7). In Figure 6 there is a good correlation in the _18O records between the SN17, Dongee and Xianshui during the 4.2 ka BP event, but for the other dates the fluctuations vary considerably. How can one examine the effect that different spatial distribution of precipitation could have on this environment.

Answer: The spatial distribution of precipitation in monsoonal China is caused by the rainbelt shifts derived from the intensity variations in East Asian summer monsoon, because the northward migrations of the rainbelt are characterized by two discontinuous jumps when summer monsoon increases. Similar to your fifth question, we will discuss the spatial distribution of precipitation during 4.2 ka BP event and its influencing factors in the revision.

(8). In Figure 7.H the _13C values for SN17 should be displayed according to the header underneath. The scale seems to be wrong and should be corrected.

Answer: For figure 7H, the labeling of Y-axis should be $\delta13C$, the values are right.

(9). In Figure 7 _18O concentration on C speleothem is lesser indicating intensified monsoon at the 4.2 ka BP event. In contrast to A,C, D where the concentration has higher values. C and D are very near and in a slightly lower latitude whereas A is slightly westwards. What would be the explanation for these values given the spatial connection of those speleothems?

Answer: Similar to your fifth and seventh questions, we will discuss the spatial distribution of precipitation and its influencing factors during the 4.2 ka BP event in the revision.

---

## Author Response (AR1)

Institute of Global Environmental Change
Xi'an Jiaotong University
28 Xianning West Road, Xi'an 710049, Shaanxi, P.R. China
Tel: 86-29-83395130, Fax: 86-29-83395100

**November 13, 2018**

Dear Dr. Zanchetta,

On behalf of my co-authors, I am resubmitting our manuscript entitled "**Hydroclimatic variations in southeastern China during the 4.2 ka event reflected by stalagmite records**". We would like to express our gratitude to the insightful comments and the suggestions by the reviewers. We also very appreciated your consideration for our manuscript. We have revised and uploaded our manuscript that incorporates the reviewers' comments and one short comment. A point-to-point response and a list of revised changes are appended with this letter. The significantly revised changes in the manuscript are highlighted in blue. All authors have read and approved this manuscript.

We thank you in advance for your consideration of this submission.

Sincerely,
Haiwei Zhang
Assistant Professor, Xi'an Jiaotong University
**99 Yanxiang road, Yanta zone, Xi'an, 710054, China**
Tel: +8615829208483
E-mail: zhanghaiwei@xjtu.edu.cn

[Figure]

General comments:

The heterogeneity of the 4.2 ka BP climatic event requires intensive researches of high-quality, high-resolution proxy records from climatically sensitive and geographically representative regions in order to reveal the spatiotemporal pattern of the event and the associated mechanism. This manuscript provided the East Asian summer monsoon with a new stalagmite record spanning the interval of 5.3-3.6 ka BP from a cave in southeast China where is a key gap of high-resolution climate records in the Asian monsoon region and investigated the possible north-south pattern of the monsoon precipitation during the 4.2 ka BP event based on the comparison of previously published proxy records from southern and northern China. The data and inferences presented in the study are of great significance and would contribute to a better understanding of the mechanism responsible for East Asian summer monsoon variations on millennial to centennial scales. I recommend acceptance of this manuscript for publication in CP after revisions.

1. The manuscript interpreted the newly obtained stalagmite record and depicted the process of East Asian summer monsoon changes during the interval of 5.3-3.6 ka BP. Although it is necessary to do so, the 4.2 ka BP event itself should be paid more attention given that the manuscript is expected to contribute to the Special Issue "The 4.2 ka BP climatic event". I understand the authors' inference that the 4.2 ka BP event might manifest a wet spell in southern China but a dry spell in northern China. What is the timing of the 4.2 ka BP event occurring in monsoon China? When did it start and end in southern and northern China, respectively? Where does the boundary lies if the event displays different regional manifestations in northern and southern China? I suggest that the authors give more discussions about these issues.

*Answer# Thank you very much for these suggestions. In the revision lines 269-323 we focus on the discussion of the timing and structure of the 4.2 ka BP event. We discuss the start and end timing of the 4.2 ka BP event in northern and southern China, and provide the boundary between the dry north and the wet south.*

2. The authors made a comparison between stalagmite records and peat ones to investigate the spatial manifestation of the 4.2 ka BP event in the monsoon region. As everyone knows, peat sequences are unparalleled in both dating precision and resolution with stalagmites. In view of the sufficient number of the published stalagmite records from the monsoon China, I suggest that the authors remove the peat records mentioned in the manuscript and focus on the existing stalagmites records.

*Answer# The peat records were removed from the comparison (figure 7), however, we still use them in the discussion to better understand the timing and nature of the 4.2 ka BP event in monsoonal China and to locate the boundary between the dry north and the wet south (figure 8).*

Specific comments:

1. Abstract on lines 19-32. Better to clearly explain the nature, especially the timing of the 4.2 ka BP event in southern and northern China.

*Answer# Lines 19-22 were revised.*

2. Lines 98-105. Is it possible to go over 1.5 km from the cave entrance to get stalagmites that consist of pure calcite? It is worthwhile if possible, because the cave lies in a key zone in monsoon China as shown in Figure 1.

*Answer# Yes, the calcite speleothems can be found in the more distal parts of the cave. We have*

*already described this in lines 99-101. This was also discussed in another paper by Zhang et al. (2015) which is now cited in line 109.*

3. Lines 117-119. State the purpose of sampling in different thickness intervals.

*Answer# 0-75 mm from the top was deposited between 3.7 and 4.6 ka BP with a low growth rate. A higher resolution sampling of this section might show more detailed information around 4.2 ka BP.*

4. Lines 134-154. This paragraph, as a part of the results, should be focused on the description of features of _18O and _13C fluctuations on different timescales during the study interval. Remove the part regarding discussions of isotopic equilibrium (lines 140-151) to the next paragraph "4.1 Interpretation of _18O and _13C".

*Answer# In lines 147-160 the features of the $\delta^{18}O$ and $\delta^{13}C$ records are described in detail. Lines 140-151 were moved to section 5.1. In lines 165-182 (section 5.1) we test the isotopic equilibrium of the stalagmite from Shennong Cave.*

5. Lines 158-169. Delete or reduce this part.

*Answer# Was reduced.*

6. Lines 169-178. Show the location of E'mei cave in Figure 1, and add one Figure to show the correlation between the speleothem _18O record from E'mei cave and the EASM precipitation amount in 1951-2009 AD.

*Answer# We show the location of E'mei cave in Figure 1. In our paper (Zhang et al., 2018) figure 3 shows the correlation between the E'mei $\delta^{18}O$ record and the precipitation amount for 1951-2009AD.*

7. Lines 179-192. Reduce this part and consider to integrate this part with the part on lines 140-151 to briefly explain 1) the relation between _18O and _13C (isotopic equilibrium), and 2) the implications of _18O and _13C.

*Answer# These two parts were reorganized into sections 5.1 and 5.2.*

8. Lines 205-252. This paragraph should be organized only on the basis of the data obtained from this study. Remove lines 225-227 to "4.3". Delete lines 245-249. Remove lines 249-252 to "4.3". More importantly, rewrite a new paragraph on the basis of the part on lines 242-244 to explain the nature and timing of the 4.2 ka BP event reflected by the study stalagmite.

*Answer# This paragraph was reorganized into section 5.3.*

9. Lines 254-307. Delete the first paragraph on lines 255-271. The part "4.3" should give a clear view of 1) the nature and timing of the 4.2 ka BP event in southern and northern China, and the boundary between the dry north and the wet south based on the comparison of stalagmite records from monsoon China (eastern China).

*Answer# In section 5.4 we discuss the nature and timing of the 4.2 ka BP event in monsoonal China, and the boundary between the dry north and the wet south.*

10. Figure 1. Remove sites of the peat records from Panel A, and show SN in a different sign.

*Answer# Figure 1 was redrawn according to ths suggestion.*

11. Figure 2. Remove "0" from the X axis of Panels A and B and show months consecutively (better as abbreviations in English). In Panel C, the tick marks for each time interval showing on the X axis seem to be one less.

*Answer# Figure 2 was redrawn according to these suggestions.*

12. Figure 7. Delete Panel E. Panel H should be _13C rather than _18O.

*Answer# Figure 7 was redrawn according to this suggestion.*

13. Table 1. Better to show the distance from the top for each sample.

*Answer# A column showing the distance from the top for each sample was added.*

Related aspects: 1. Does the paper address relevant scientific questions within the scope of CP? Yes. 2. Does the paper present novel concepts, ideas, tools, or data? Yes. 3. Are substantial conclusions reached? Yes. 4. Are the scientific methods and assumptions valid and clearly outlined? Yes. 5. Are the results sufficient to support the interpretations and conclusions? Yes. 6. Is the description of experiments and calculations sufficiently complete and precise to allow their reproduction by fellow scientists (traceability of results)? Yes. 7. Do the authors give proper credit to related work and clearly indicate their own new/original contribution Yes. 8. Does the title clearly reflect the contents of the paper? Yes. 9. Does the abstract provide a concise and complete summary? Not sufficient. 10. Is the overall presentation well structured and clear? Not sufficient. 11. Is the language fluent and precise? Yes. 12. Are mathematical formulae, symbols, abbreviations, and units correctly defined and used? Yes. 13. Should any parts of the paper (text, formulae, figures, tables) be clarified, reduced, combined, or eliminated? Yes. 14. Are the number and quality of references appropriate? Yes. 15. Is the amount and quality of supplementary material appropriate? Yes.

*Answer# Thank you very much for these critical comments and helpful suggestions. We revised the manuscript according to these suggestions.*

**Response to referee #2**

General Comments: The manuscript describes a new, well-dated isotopic record of environmental change from the transitional period between the Middle and Late Holocene. Overall, the data appear high quality and collected/analyzed properly. The presentation of the material and interpretation of these data is generally good, but does require some additional thought and discussion. Some structure and figure design changes are needed, as indicated in specific and technical comments below. Generally, this review agrees with comments made by reviewer 1, although this reviewer believes the peat record to still be of importance as it offers evidence of replicated environmental change in a different proxy, despite chronological limitations. This reviewer believes that after restructuring (detailed well by reviewer 1) and addressing the concerns below (particularly about isotopic interpretations of wet/dry), the manuscript will be acceptable for publication.

Specific Comments:

44: 'specific level' is unclear/vague. Do you mean a specific physical level (like a layer at a depth) or a geochemical threshold?

*Answer# Now explained in lines 46-48.*

47: Be careful about assigning direct causality between the climate changes and societal responses. For some better studied sites, a direct impact of climate change leading to societal collapse may be well-established enough to confidently state such. However, for many others it may be more accurate to highlight that the climate and societal changes coincided and were likely associated, but not certainly proven. It is also important to acknowledge that climate-societal interactions are usually much more complex than our simplified paleo-perspectives (e.g., there was drought, so therefore their society suffered and collapsed).

*Answer# We changed this expression in lines 49-50.*

76: Data source for Guixi data?

*Answer# Now included.*

77: A sentence further explaining the climatic set up and characteristics of the spring persistent rainfall would help here, since it seems to be an interesting and important regional characteristic
82: "Data from..." sentence is convoluted and difficult to read as is.

*Answer# In lines 78-96, the climatic setup and characteristics of the spring persistent rainfall is explained. The sentence "Data from…" was also rewritten.*

87: Is it important that it was found after days of heavy rain? Was it previously not open/accessible?

*Answer# No, we just described how this cave was found. Before the heavy rain in 1998, nobody knew this cave. This sentence was deleted.*

92: You have some taxonomic inconsistency reporting plants here: Pinus is a genus, Taxodiaceae is a family. "Camelliaoleifera" should be a binomial genus/species: Camellia oleifera. Bamboo is only given as a common name. Preferably, you should list plants on the same taxonomic level (probably just genus), and species level is probably not necessary for your discussion here. Also, Taxodiaceae is no longer a recognized plant family; it has since been absorbed into Cupressaceae.
*Answer# In lines 100-101 these plant names were corrected.*

129: Was evidence of hiatus examined petrographically? Or is this conclusion simply based on the age distribution? The top 50 mm have a few petrographic boundaries I can see in Fig 3 that might be worth examining closer for short hiatuses petrographically (if you haven't already done so) (e.g., Railsback 2013).

*Answer# We did not check the petrography; there might be two short hiatuses at 10 and 50 mm. Thin sections cannot be done in our lab, and we did not have enough time. We increased the number of U-Th ages above and below these two layers to check for hiatuses, but the new dates and the COPRA-based age model indicate that there is no significant hiatus.*

131: The linear age-depth model looks sufficient. It might be worthwhile to age model with BACON or StalAge and see if that changes any results/interpretation significantly.

*Answer# We used COPRA model.*

150: It may be useful to note that even stalagmites that deposit with kinetic fractionation can still preserve valuable climate data BECAUSE of the fractionation. So even if your stalagmite isn't in isotopic equilibrium, it can still have useful data (though your interpretation of the isotopes may be different).

*Answer# For the "Hendy test" we analyzed twenty-one subsamples from three growth layers (figure 5 and section 5.1). As confirmed by the "Hendy test" and the "replication test" (section 5.1) stalagmite SN17 was likely deposited close to isotopic equilibrium.*

165: "orbital" is not a timescale. Millennial timescales should suffice. However, on the timescale you are examining, orbital forcings are not a factor, so this is a somewhat weak/irrelevant point. Focus on what the literature says about controls on d18O for the decadal/centennial range you are examining.

*Answer# Corrected.*

171: "We suggest"- Are you suggesting that conclusion newly in this paper? Or was this the conclusion of Zhang 2018 you cite? If the latter, I would rephrase to simply state that data from E'mei cave concluded that EASM-NSM balance controls the d18O, and not say "we" concluded it.

*Answer# Corrected.*

171: Earlier (65) you said there was only one published stalagmite from SE China, but isn't Zhang 2018 another published record from SE China?

*Answer# No, this is a different paper (Zhang et al., 2004), a record from Xiangshui Cave. In line 67 this is now clearly stated.*

178: A sentence clarifying and summarizing how you are interpreting the d18O in Shennong Cave would be nice here, since you state several possible ways to interpret d18O for the region.

*Answer# In lines 188-203 several sentences were added to interpret the $\delta^{18}O$ data from Shennong Cave.*

190: Your d13C summary is generally good. Some supplemental resources you might want to examine include Oster et al., 2010; Meyer et al., 2014; Noronha et al., 2015; Wong and Breecker, 2015 to get more recent studies and summaries on d13C.

*Answer# Thanks for this recommendation - some of them are now included.*

200: I think your dismissal of the effects of degassing and PCP is premature. Some degassing must occur in order for CaCO3 precipitation to occur (being deposited in perfect 'isotopic equilibrium' is impossible, since a system in equilibrium will not undergo any reactions or change). And the presence of soda straws and stalactites (which I assume are present in the cave) means PCP is also occurring. The negative relationship between d13C and growth rate suggest to me that PCP is perhaps quite important as a control. Perhaps more importantly, you could argue that vegetation dynamics are a major or the major control on d13C, but when multiple factors are working in concert (e.g., drier conditions both lead to less vegetation and greater PCP which both lead to higher

d13C values), dismissing one or more potential factors is not even necessary.

*Answer# We do not entirely exclude the effects of degassing and PCP. In lines 220-223 we state that "Stalagmite SN17 was LIKELY deposited CLOSE to isotopic equilibrium, as confirmed by the "Hendy test" and "replication test". The $\delta^{13}C$ variations in this stalagmite were PRIMARILY driven by vegetation density and soil bioproductivity associated with hydroclimatic variations but not PCP or rapid $CO_2$ degassing......"*

214: Do you have any supporting evidence that the d18O for your stalagmite reflects annual precip (e.g., through drip water monitoring?) or is this an assumption? I think the match between it and Dongge make a decent argument that your stal is recording long-term aggregates rather than 'flashy' storm events. But how you decided that it is annual precip should be mentioned.

*Answer# Yes, we have done monitoring work in this cave for two years. The seasonal variation of drip water $\delta^{18}O$ shows very small variations (~6‰ in the whole year), consistent with the amount-weighted annual precipitation $\delta^{18}O$ value outside the cave. This is now clearly stated in lines 91-93.*

222: Wouldn't wet intervals be those with z scores less than zero? (Not greater, like you have written). Also, You earlier state that d18O is interpreted in your area as the ratio between EASM and NSM amounts, with lower values meaning a greater fraction of EASM. Shifting the seasonality of precipitation can therefore change the d18O in the stalagmite without actually changing annual precipitation amounts. Additionally, a stalagmite d18O that decreases could be because the EASM gets more intense (more overall rainfall), but also when the NSM decreases more than the EASM (less overall rainfall). Be careful about interpreting d18O as amount unless you have supporting evidence.

*Answer# This part was rewritten. In the lines 183-204 and 227-262 the $\delta^{18}O$ record is now better explained. We suggest that the $\delta^{18}O$ record from Shennong Cave might be primarily influenced by EASM/NSM ratio and also affected by the EASM precipitation amount on interannual to decadal timescales, and can be dominated by EASM precipitation amount on decadal to centennial timescales, i.e., lower (higher) $\delta^{18}O$ values corresponding to higher (lower) EASM/NSM ratios and more (less) EASM precipitation.*

223: "More wet intervals": Wouldn't a better measurement be "more years wetter than average"? This sounds like you are just counting the number of times you have a span below 0 Z-score (so a highly variable record with many changes above and below average could easily have more 'wet intervals' than a record that is all wetter than average in a single long 'wet interval').

*Answer# This section was deleted.*

227: Controlled by what variable of summer monsoon precip? Amount? EASM/NSM like your cave?

*Answer# It is the summer monsoon precipitation amount. Corrected.*

230: Growth rate is often not a direct function of precipitation amount (e.g., Railsback 2018). If you believe growth rate in your stal is a direct relationship to precip amount, some supporting evidence/arguments would be beneficial.

*Answer# This part was rewritten (lines 232-241).*

238: 150 years is a pretty long time to be a vegetational response delay in terms of vegetation coverage, particularly if there is not a significant shift in vegetation type. Do you have a more detailed explanation of why the vegetational response would take 150 years? Are there alternative reasons that could explain the lag? Perhaps d13C is showing actual precip amount changes, and the 'lag' is because the d18O can reflect proportional shifts in EASM/NSM that may not result in actual precip amount changes.

*Answer# This part was rewritten (lines 253-256).*

255: The previous paragraph contained records in monsoonal China covering the 4.2 ka event. Why are they separate from section 4.3?

*Answer# This part was reorganized and all of the records from monsoonal China are discussed in section 5.4.*

287: While the argument linking EASM intensity to AMOC is sound, the IRD record is not particularly strong evidence since the variance in IRD between 3700 and 4500 yr BP is quite small. Are there alternative records for AMOC intensity you could use, or perhaps support this by bringing in records also showing monsoonal changes in Africa and South Asia at this time.

*Answer# A stalagmite record from Mawmluh Cave in India was added for comparison and discussion.*

294: Are you calculating coherence, or do you mean the variables co-vary?

*Answer# We mean that the variables co-vary.*

Fig 1: Map A is too far zoomed out and is difficult to see sites. Ideally would have main part of map and this figure focused on eastern China. Small inset map could provide wider context. Maps also need a legend identifying icons and color scheme for basemap. Highlight your site on main map better (e.g., larger text, unique color, pointing arrow). Another map or layer on this map showing typical modern location of the summer monsoon influence/extent would be beneficial. No scale on map B. 610: You bring up several more climatic influencing winds here that are never discussed or mentioned in your paper. If they are important, they need to be discussed, or at least mentioned why you are not considering them.

*Answer# Figure 1 was redrawn according to this suggestion. The base map presents the Natural Earth physical map at 1.24 km per pixel for the world (data source: US National Park Service). Source and more information: http://goto.arcgisonline.com/maps/World_Physical_Map. It is now stated in the caption of figure 1.*

Fig 4: Labeling the y axes with the environmental interpretation (e.g., wetter/drier, more intense EASM, etc) would aid the understanding of these plots

*Answer# Redrawn.*

Fig 6: Labeling the axes with the cave name (or directly on the plot) along with in the caption would make the plot more readable. The coarse resolution of Xiangshui makes it very difficult for me to conclude anything about the covariation between it and your record. I do think the Dongge records visually matches well.

*Answer# Redrawn. The Xiangshui record was kept in figure 6 because it is the only published stalagmite record of 4.2 ka BP event in southeastern China except our record from Shennong Cave.*

Fig 7: Labeling the axes with the sample/cave/site name (or directly on the plot) along with in the caption would make the plot more readable. Also, labeling the Y-axes with the environmental interpretation (wet/dry, monsoon N-S offset, etc) will help.

*Answer# Redrawn.*

Fig 7: The yellow bars don't align well with your d18O record. Is there a reason they are offset from the low value intervals of your record?

*Answer# In revision, we focus on the timing and nature of the 4.2 ka BP event. Therefore, figure 7 was redrawn.*

Technical comments:
73: Your latitude/longitude is flipped

*Answer# Corrected.*

81: Shennong Cave or Shennong cave? Capitalization consistency. Also, this sentence seems unnecessary and out of place as you already mentioned that the cave is in the region of spring persistent rainfall.

*Answer# We checked the whole manuscript and corrected them.*

160: Re-examine your use of commas in this sentence. It's unclear which phrases are meant to be grouped in the list of influences.

*Answer# Rewritten.*

188: Prior not needed to be capitalized

*Answer# Corrected.*

196: Cave or cave? I think that Cave should be used when referring to specific named caves here and throughout, but it's more important for you to be consistent with capitalization.

*Answer# Corrected.*

211: 'wetter to drier conditions' is better, because there wasn't a major regime shift into definitively 'dry' conditions from earlier 'wet' conditions

*Answer# Corrected.*

255: "A remarkable drought" is better here than "the remarkable drought", since you haven't discussed the drought for the past few pages.

*Answer# Corrected.*

268: "the large dating uncertainties and the low resolution" Change to "by large dating uncertainties and low proxy resolution in many records"? or something more clear

*Answer# Corrected.*

610: Westerly used here sounds like you are saying westerly monsoon, but you are probably just referring to the westerlies, correct?

*Answer# Corrected.*

Figure 2: You may wish to recolor the portion with the red-green lines. Almost 1 in 10 people suffer from some degree of red-green colorblindness.

*Answer# Redrawn.*

Fig 3: Just a design though: Your age markers are red on the plot, but black on the stalagmite. The red marks on the stalagmite are XRD. For consistency and ease of eyematching of this figure, you might consider making the age markers on the stalagmite red and the XRD markers a different color.

*Answer# Redrawn.*

Figure 4: You may wish to recolor the portion with the contrasting red-green Z score. Almost 1 in 10 people suffer from some degree of red-green colorblindness.

*Answer# Redrawn.*

**Response to short comment by Charilaos Tziavaras**

The purpose of this paper is to examine the link between environmental changes in the 5.3 and 3.6 ka BP period and the collapse of several Neolithic cultures in China like the Shijiahe Culture which was located in the middle reaches of the Yangtze River. Bigger focus is given on the 4.2 ka BP event during which dryer conditions insisted on the northern parts of China compared to the southern part, where the conditions where more wet. This is done by reconstructing regional monsoon intensity from a stalagmite in Shennong Cave (SN17). The reconstruction is done through _18O and _13C proxy analysis. The results are cross-referenced with other proxy data from previous work done on China monsoon. The explanation claimed for these climatic changes on the 4.2 ka BP event are attributed to a weaker East Asian summer monsoon (EASM) due to reduced Atlantic Meridional Overturning Circulation (AMOC) which led to a southward migration of the Intertropical Convergence Zone. Paleoenvironmental reconstruction research to help us understand about the conditions surrounding the development of past civilizations and cultures is very important. This paper can give another aspect on this research front, with a greater focus on Southeastern Asia.

Comments.
(1). The methods chapter was very simply put with references provided so if readers are further interested they could further examine the details of techniques used.

*Answer: Yes, the related references were provided. Because the methods of U-Th dating and stable isotope measurement are extensively used in speleothem and paleoclimatic studies, the readers can*

*find the similar descriptions in many papers.*

(2). This paper's results can give further information on how the EASM intensity can be derived from _18O concentration in speleothems.

*Answer: We discussed the significance of the speleothem $\delta^{18}O$ from the region of spring persistent rain in southeastern China. According to two reviews´ suggestion speleothem $\delta^{18}O$ data from Shennong Cave are now better explained in the revised section 5.2.*

(3). Are there enough evidence which support that ice-rafted debris decreased AMOC intensity during that period?

*Answer: AMOC might be a possible reason. In the published papers, such as Wang et al. (2001), Tan et al. (2011, 2018), Chiang et al. (2015), and Railsback et al. (2018), a reduced AMOC was considered as a possible reason for a southward migration of the ITCZ and a weakened East Asian summer monsoon. The weakened monsoon during the 4.2 ka BP event corresponds to higher amounts of ice-draft debris in the North Atlantic (Figure 7I). The CPD paper Yan et al. (2018), part in the special issue on the 4.2 ka BP event, discusses the possible reason for the 4.2 ka BP event using a set of long-term climate simulations. In this paper, all forcing experiments show that the 4.2 ka BP event could have been related to the slowdown of the AMOC, and the comparison between the all-forcing experiments and the single-forcing experiments indicates that the event was likely caused by internal variability. This is now discussed in lines 324-354.*

(4). E'mei cave is being mentioned on 171 line without the being labeled on Figure 1 where all the other sites are recorded. Since the regional environmental changes have a spatial significance it would be appropriate if it was presented.

*Answer: The location of E'mei Cave is shown on Figure 1.*

(5). Various parameters are presented that are able to influence the _18O of the paper's speleothem. It would be useful that the conditions of these parameters were presented for the other speleothems used in this paper to compare with the data acquired by the authors. Also to this note it may be good if the paper urges other researches into further investigating the reasons behind the amplitude difference between the SN17 and Jiuxian and Xianglong speleothems.

*Answer: According to the second reviewer's suggestion and yours, speleothem $\delta^{18}O$ data from Shennong Cave are now better explained in revised section 5.2. In section 5.4 we discuss the timing and nature of the 4.2 ka BP event in northern and southern China by comparison with different speleothem $\delta^{18}O$ records from monsoonal China.*

(6). Shennong Cave is located in an area that is influenced also from spring persistent rain. Maybe this is something that needs further investigation since it provides the area with a surplus of water compared to the more naturally dry northern part of China. Do the authors think that this is something worth considering?

*Answer: In lines 78-96, the climatic setup and characteristics of the spring persistent rainfall is explained. In lines 185-207, speleothem $\delta^{18}O$ data from Shennong Cave are now better explained. To sum up, in the region of spring persistent rain the EASM (May to September) precipitation accounts from 54% of the annual precipitation and the non-summer monsoon (NSM, October to next April) precipitation accounts for 46%. The distribution of EASM vs. NSM precipitation amount in this region is distinctly different from that in the northern and southwestern part of monsoonal*

*China, where the mean percentage of EASM/annual (65-90%) is much higher than the mean percentage of NSM/annual (10-35%). Therefore, we suggest that the stalagmite $\delta^{18}O$ record from Shennong Cave might be primarily influenced by the EASM/NSM ratios and also affected by the EASM precipitation amount on interannual to decadal timescales, and can be dominated by EASM precipitation amount on decadal to centennial timescales, i.e., lower (higher) $\delta^{18}O$ values corresponding to higher (lower) EASM/NSM ratios and more (less) EASM precipitation. Detailed discussions can be found in lines 78-96 and 183-203.*

(7). In Figure 6 there is a good correlation in the _18O records between the SN17, Dongee and Xianshui during the 4.2 ka BP event, but for the other dates the fluctuations vary considerably. How can one examine the effect that different spatial distribution of precipitation could have on this environment.

*Answer: The spatial distribution of precipitation in monsoonal China might be caused by shifts of the rainbelt due to intensity variations of East Asian summer monsoon, because the northward migration of the rainbelt is characterized by two discontinuous jumps when the summer monsoon increases. We discuss the spatial distribution of precipitation and its influencing factors in lines 324-354.*

(8). In Figure 7.H the _13C values for SN17 should be displayed according to the header underneath. The scale seems to be wrong and should be corrected.

*Answer: Corrected.*

(9). In Figure 7 _18O concentration on C speleothem is lesser indicating intensified monsoon at the 4.2 ka BP event. In contrast to A, C, D where the concentration has higher values. C and D are very near and in a slightly lower latitude whereas A is slightly westwards. What would be the explanation for these values given the spatial connection of those speleothems?

*Answer: Similar to your seventh question, we discuss now the spatial distribution of precipitation and its influencing factors in lines 324-354.*

**List of revised changes**

**Lines 83-96**

[revised manuscript text omitted]

We also added one more figure (Fig. 8) to show the spatial distribution of precipitation and the boundary between northern and southern China during the 4.2 ka BP event. The related references were also cited.